# KDM3 epigenetically controls tumorigenic potentials of human colorectal cancer stem cells through Wnt/β-catenin signalling

Jiong Li[1], Bo Yu[1], Peng Deng[1], Yingduan Cheng[1], Yongxin Yu[1], Kareena Kevork[1], Sivakumar Ramadoss[1], Xiangming Ding[2], Xinmin Li[2] & Cun-Yu Wang[1,3]

Human colorectal cancer stem cells (CSCs) are tumour initiating cells that can self-renew and are highly tumorigenic and chemoresistant. While genetic mutations associated with human colorectal cancer development are well-known, little is known about how and whether epigenetic factors specifically contribute to the functional properties of human colorectal CSCs. Here we report that the KDM3 family of histone demethylases plays an important role in tumorigenic potential and survival of human colorectal CSCs by epigenetically activating Wnt target gene transcription. The depletion of KDM3 inhibits tumorigenic growth and chemoresistance of human colorectal CSCs. Mechanistically, KDM3 not only directly erases repressive H3K9me2 marks, but also helps to recruit histone methyltransferase MLL1 to promote H3K4 methylation, thereby promoting Wnt target gene transcription. Our results suggest that KDM3 is a critical epigenetic factor in Wnt signalling that orchestrates chromatin changes and transcription in human colorectal CSCs, identifying potential therapeutic targets for effective elimination of CSCs.

[1] Laboratory of Molecular Signaling, Division of Oral Biology and Medicine, School of Dentistry and Broad Stem Cell Research Center, UCLA, Los Angeles, California 90095, USA. [2] Department of Pathology and Laboratory Medicine, David Geffen School of Medicine, UCLA, Los Angeles, California 90095, USA. [3] Department of Bioengineering, Henry Samueli School of Engineering and Applied Science, UCLA, Los Angeles, California 90095, USA. Correspondence and requests for materials should be addressed to J.L. (email: jli@dentistry.ucla.edu) or to C.-Y.W. (email: cunywang@ucla.edu).

Wnt/β-catenin signalling controls important biological processes including normal development, stem cell self-renewal and differentiation, and oncogenesis[1–5]. The hyperactivated Wnt/β-catenin signalling pathway has been found to be associated with various types of human cancers, most notably colorectal cancers (CRCs) due to *APC* and *CTNNB1* (β-catenin) mutations[2,6,7]. In these cases, Wnt/β-catenin signalling promotes oncogenesis by inducing the expression of Wnt target genes such as Cyclin D1 and c-Myc. In the absence of β-catenin, Wnt target genes are silenced by the T cell factors (Tcfs) and their transcriptional corepressors such as Groucho/transducin-like enhancer protein 1 and histone deacetylase 1 (refs 8–10). To activate transcription, β-catenin needs to replace Groucho/transducin-like enhancer protein 1 from Tcf through competitive binding and recruit co-activators and chromatin-remodeling complexes[5]. The transcriptional co-activators, including CBP/P300, B-cell lymphoma 9 (BCL9)/Pygopus (PYGO), polymerase-associated factor 1 and SET1 have all been reported to interact with β-catenin during transcriptional activation[11–15].

Colorectal cancer is the third most common cancer worldwide and the fourth most common cause of death[16]. A small subset of cancer stem cells (CSC), or cancer initiating cells with the ability to self-renew and maintain the tumour, have been isolated from human CRCs. Numerous reports have highlighted the importance of Wnt/β-catenin signalling in CSC self-renewal and oncogenesis[17–19]. The CSC model is also implicated in tumour recurrence and development of drug resistance. Because of the intrinsic stem cell-like properties of CSCs, this small percentage of tumour cells cannot only initiate and maintain tumour growth but also develop resistance to chemotherapy[20]. *APC* and *CTNNB1* (β-catenin) mutations are the major cause of the abnormal activation of Wnt/β-catenin signalling in human CRCs. Interestingly, hyperactivated Wnt/β-catenin signalling has been shown to be an important characteristic of CSCs in human CRCs[21–23]. Therefore, understanding Wnt/β-catenin signalling in CSCs might help to develop novel targeting strategies for eliminating CSCs, thereby improving the clinical outcomes of patients with CRCs.

Histone methylation plays a critical role in controlling gene transcription by altering chromatin accessibility[24–26]. Emerging evidence suggests that epigenetic factors might help to govern colon tumour initiation. Although *APC* and *CTNNB1* mutations play a critical role in human CRC development, epigenetic and genetic alternations are likely to act synergistically in human CRC development. While H3K4 methylation is critical for gene activation, H3K9 and H3K27 methylations are associated with gene silencing[26]. Bivalent chromatin domains, characterized by co-existence of both active H3K4me3 and repressive H3K27me3 marks, have been found to play an important role in regulation of gene expression in both embryonic stem cells and adult stem cells[27–30]. H3K4me3 is important for the expression of Wnt target genes by facilitating chromatin association with the co-activators PYGO2 and BCL9 (refs 31–33). Interestingly, loss of H3K27me3 from bivalent promoters was found to accompany the activation of genes associated with human CRC progression and CSC phenotype, suggesting that chromatin architecture in CSCs might be different from that in embryonic stem cells[34,35]. However, whether modifying H3K9 methylation regulates human colorectal CSCs is still unclear. As the hyperactivated Wnt/β-catenin-mediated transcription activities define the CSC phenotype[21–23], elucidating the underlying epigenetic mechanisms that control Wnt target gene transcription might have important implications in developing novel therapeutic strategies for effective elimination of CSCs. A group of histone demethylases activate or inhibit gene transcription by removing histone methylation marks. Histone demethylases catalyse oxidative demethylation reactions with iron and α-ketoglutarate as cofactors[24]. To explore whether and how epigenetic factors interact with transcription factors to control the CSC phenotype, we performed a functional *in vitro* siRNA screen to identify potential histone demethylases that may regulate β-catenin/Tcf-dependent transcription.

Our screen reveals that the KDM3 family histone demethylases play a critical role in the oncogenic potential of CSCs by controlling Wnt/β-catenin-mediated transcription. The KDM3 family histone demethylases, including KDM3A, KDM3B and JMJD1C, can remove the methyl groups from H3K9me2 to activate target gene expression[36,37]. In addition to erasing H3K9me2 marks, we find that that KDM3 facilitate PYGO2 and BCL9 binding to chromatin by histone methyltransferase MLL1-mediated H3K4 methylation. Moreover, the depletion of KDM3 significantly inhibits tumorigenic potentials and survival of CSCs.

## Results

**KDM3 is required for Wnt/β-catenin-mediated transcription.** We performed a functional siRNA screen to identify the potential histone demethylases required for β-catenin/Tcf-mediated transcription. We found that the knockdown of the KDM3 family histone demethylases inhibited Topflash luciferase activities induced by LiCl in human embryonic kidney 293 T cells expressing a Tcf-responsive Topflash reporter (293 T/Top) (Fig. 1a). The KDM3 family demethylases contain three subfamily members, namely KDM3A, KDM3B and JMJD1C, which are responsible for removing the methyl groups from H3K9 with a preference for H3K9me2. Efficient knockdown of KDM3A, KDM3B and JMJD1C was confirmed by Real-time RT-PCR (Fig. 1b) and western blot (Fig. 1c). To further determine whether KDM3 family members were required for β-catenin/Tcf-mediated transcription, we also examined whether their knockdown impaired the expression of the well-known Wnt target genes *AXIN2* and *DKK1*. Depletion of KDM3A or KDM3B led to over 50% inhibition of the expression of *AXIN2* and *DKK1* induced by LiCl, while depletion of JMJD1C weakly affected the expression of *AXIN2* and *DKK1* (Fig. 1d). Simultaneous triple depletion of KDM3A, KDM3B and JMJD1C or double depletion of KDM3A and KDM3B had a similar effect and almost completely abolished the expression of *AXIN2* and *DKK1* (Fig. 1d). This indicates that JMJD1C likely plays a minor role in β-catenin/Tcf-mediated transcription. Our results also suggest that there was a functional redundancy between the KDM3 family members. Therefore, we simultaneously knocked down both KDM3A and KDM3B (KDM3A/B) in most of our studies. The depletion of KDM3A/B also inhibited the expression of *AXIN2* and *DKK1* induced by Wnt3a, a canonical Wnt ligand (Fig. 1e).

Overexpression of KDM3A or KDM3B significantly enhanced the Topflash reporter activation induced by a mutant form of β-catenin in a dose-dependent manner (Fig. 1f). To rule out off-target effects of the siRNA, we transiently transfected siRNA-resistant KDM3A in KDM3A/B-depleted 293 T cells. The ectopic expression of KDM3A fully restored Topflash reporter activities (Fig. 1g) and the expression of *AXIN2* and *DKK1* induced by Wnt3a (Fig. 1h). To further delineate whether KDM3 histone demethylase activity is required for β-catenin/Tcf-mediated transcription, we generated a mutant KDM3A-H1120A in which the demethylase activity was abolished[36]. Overexpression of KDM3A-H1120A could not enhance Topflash reporter activities induced by β-catenin (Fig. 1i) or Wnt3a (Fig. 1j).

Because Wnt/β-catenin signalling plays an important role in human CRC development, we examined whether KDM3 epigenetically regulated Wnt target genes in human colorectal cancer cells. We utilized shRNA to stably knockdown KDM3A/B

expression in a human colorectal cancer cell line HCT116. HCT116 cells have constitutively active β-catenin/Tcf-mediated transcription due to Ser45 mutation in β-catenin. To rule out off-target effects, we utilized two different shRNAs targeting KDM3A/B (ABsh-1 and ABsh-2). Western blot analysis

confirmed efficient knockdown of KDM3A/B by ABsh-1 and ABsh-2 (Fig. 1k). Real-time RT-PCR showed that both ABsh-1 and ABsh-2 significantly inhibited the expression of well-known Wnt target genes, including *AXIN2*, *DKK1*, *CCND1* and *MYC* (Fig. 1l). We then performed microarray analysis on

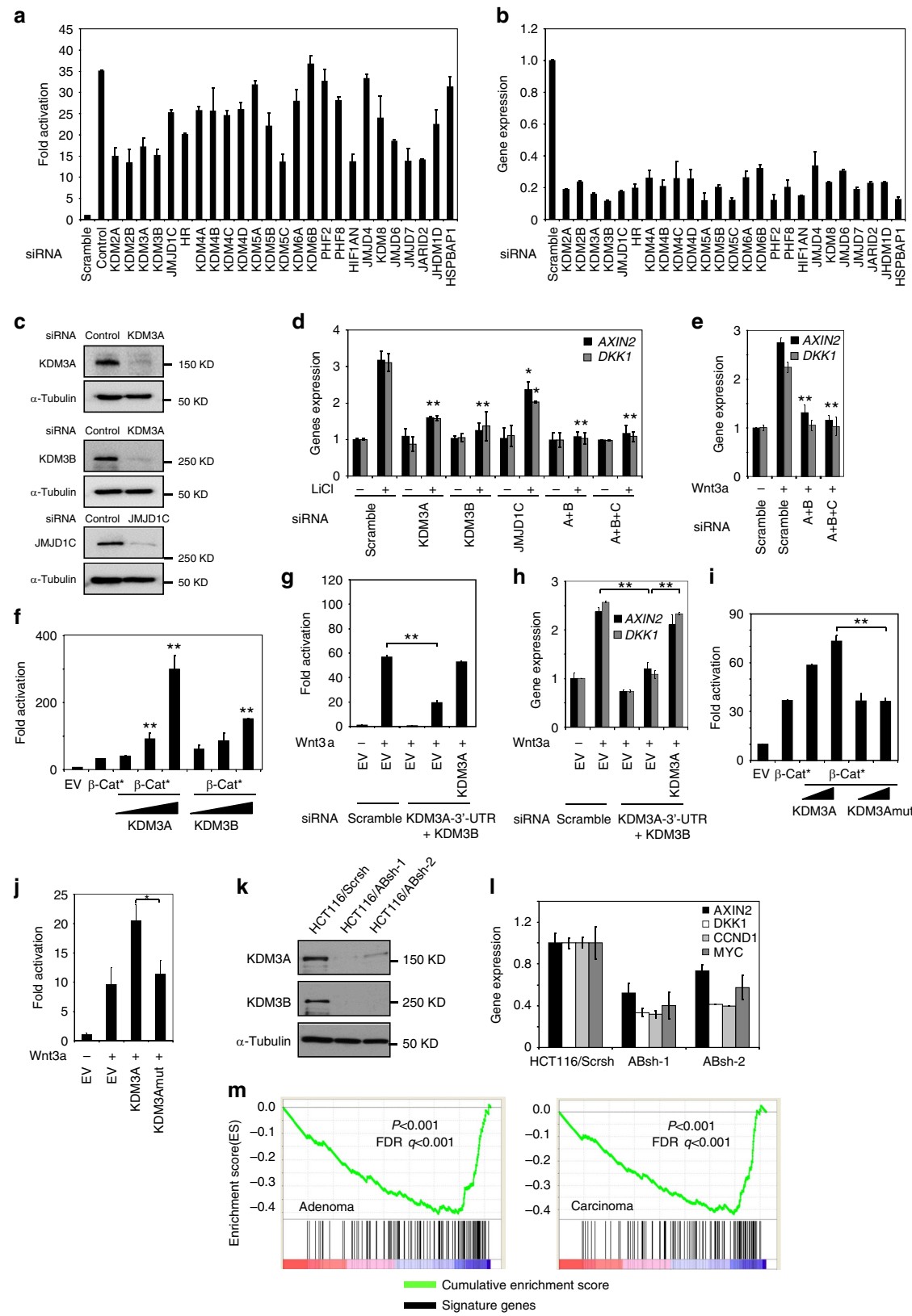

HCT116/Scrsh and HCT116/ABsh-1 cells to examine the effect of KDM3A/B depletion on the global gene transcriptome. Compared with published gene expression profiles from CRC cell lines[38,39], gene set enrichment analysis (GSEA) revealed that, there existed a significant negative correlation between Wnt/β-catenin upregulated genes and downregulated genes following KDM3A/B depletion, highlighting the critical role of KDM3 in β-catenin/Tcf-mediated transcription (Fig. 1m). To further examine the role of the KDM3 family demethylases in human CRC development *in vivo*, we examined the expression of these demethylases in colorectal tumours and matched adjacent normal tissues using a human colon cancer tissue microarray (TMA; Supplementary Fig. 1a). Consistent with our previous study[40], we found that over 90% of TMA samples were positively stained for cytoplasmic and nuclear β-catenin (Supplementary Fig. 1a). Wilcoxon signed rank test indicated that the expression of KDM3A and KDM3B were significantly higher in the human CRC tissues than in the adjacent normal tissue ($P<0.001$) (Supplementary Fig. 1c,d; Supplementary Table 1).

**KDM3A/B directly interact with β-catenin.** Since the histone demethylases can be recruited by various transcriptional activators[36,41,42], we next examined whether KDM3 family demethylases could interact with β-catenin, the key transcriptional activator of canonical Wnt pathway. Co-immunoprecipitation (co-IP) assays in 293 T cells showed the interaction between HA-tagged β-catenin and Flag-tagged KDM3A (Fig. 2a). A similar interaction was observed between Flag-tagged β-catenin and HA-tagged KDM3B (Fig. 2b). To explore whether endogenous β-catenin interacts with KDM3 family demethylases to regulate Wnt signalling, we performed a co-IP assay using antibodies against endogenous KDM3A and KDM3B in the nuclear extract of 293 T cells. On Wnt3a stimulation in 293 T cells, there was a noticeable increase in the interaction between β-catenin and KDM3A/B (Fig. 2c). Similarly, nuclear co-IP assays in the human colorectal cancer cell line HCT116 verified the interaction between endogenous β-catenin and KDM3A or KDM3B (Fig. 2d). To examine whether β-catenin could directly interact with KDM3A and KDM3B, we generated an *in vitro* translated KDM3A and KDM3B. GST-pull down assays revealed that the recombinant glutathione S-transferase fusion β-catenin could directly interact with KDM3A and KDM3B (Fig. 2e).

**KDM3 maintains tumorigenic potential of CSCs in human CRCs.** Wnt/β-catenin plays a critical role in the development and progression of human CRC. In addition, it has been reported that high Wnt/β-catenin activities play a critical role in maintaining the CSC self-renewal and tumorigenic potential[21,22]. Therefore, we were interested in whether KDM3A/B might epigenetically affect CSC functions by controlling Wnt/β-catenin signalling. Aldehyde dehydrogenase-positive (ADLH$^+$) subpopulations in human CRC cells exhibit CSC-like properties *in vitro* and *in vivo*[43–45]. To explore whether KDM3A/B play a critical role in maintaining CSC-like properties, we transiently depleted KDM3A/B in HCT116 cells using siRNA. Interestingly, the cell sorting profile revealed that the depletion of KDM3A/B dramatically decreased the proportion of the ALDH$^+$ subpopulation in HCT116 cells (Supplementary Fig. 2a). In agreement with this result, the ALDH$^+$ subpopulation was dramatically increased by overexpression of KDM3A/B in HCT16 cells (Supplementary Fig. 2b). To further confirm our results, we isolated ALDH$^+$ cells from HCT116 cells and then knocked down KDM3A/B with siRNA immediately after isolation. After 24 h of siRNA transfection, the cell sorting profile suggested that depletion of KDM3A/B dramatically enhanced the ALDH$^+$ to ALDH$^-$ differentiation in the ALDH$^+$ HCT116 cells (Supplementary Fig. 2c). Real-time RT-PCR also indicated that the depletion of KDM3A/B significantly inhibits the expression of Wnt target genes in ALDH$^+$ HCT116 cells (Fig. 3b). As an *in vitro* measure of CSC-like behaviour, tumorsphere formation assays were utilized as a surrogate for CSC-like self-renewal[46]. Notably, while ALDH$^+$ HCT116 cells effectively formed sphere-like colonies as opposed to ALDH$^-$ cells, knockdown of KDM3A/B dramatically abolished the tumorsphere formation ability of ALDH$^+$ HCT116 cells (Fig. 3c,d). Similarly, knockdown of KDM3A/B also significantly reduced the ALDH$^+$ subpopulation in SW480 cells (Supplementary Fig. 2d). The depletion of KDM3A/B also significantly reduced the expression of Wnt target genes (Supplementary Fig. 3f), and abolished the tumorsphere formation ability of ALDH$^+$ SW480 cells (Supplementary Fig. 3g). To determine whether KDM3A/B played an important role in the tumorigenesis of ALDH$^+$ HCT116 cells *in vivo*, we knocked them down in ALDH$^+$ HCT116 cells using shRNA. ALDH$^+$ HCT116 cells expressing scramble shRNA (ALDH$^+$ HCT116/Scrsh) and ALDH$^+$ HCT116 cells expressing ABsh-1 (ALDH$^+$ HCT116/ABsh-1) were subcutaneously inoculated into nude mice. While ALDH$^+$ HCT116/Scrsh cells rapidly formed large tumours after 3 weeks, ALDH$^+$ HCT116/ABsh-1 cells formed significantly smaller tumours based on their size and weight (Fig. 3f,g).

To further determine whether KDM3A/B controlled tumorigenic potential of CSCs, we isolated CSCs from human CRC tissues (patient case #1). We were able to isolate CSCs from primary human CRC tissues and confirm their tumorigenic potential. However, to obtain sufficient CSCs for our functional

**Figure 1 | siRNA screening of histone demethylases required for Wnt/β-catenin-mediated transcription.** (**a**) Analysis of luciferase activity of 293 T/Top cells treated with the indicated siRNA relative to those transfected with scramble siRNA. The relative luciferase activity was normalized against the protein concentration of each cell lysate sample. Values mean ± s.d. from three independent experiments. (**b**) Real-time RT-PCR analysis of the relative expression of histone demethylases in human of 293 T/Top cells transfected with targeted siRNA relative to their expression in cells with scramble siRNA. Values are mean ± s.d. of three independent experiments. (**c**) KDM3A, KDM3B and JMJD1C were knocked down by siRNA in 293 T cells. (**d,e**) Knockdown of endogenous KDM3A, KDM3B and JMJD1C inhibited *AXIN2* and *DKK1* expression induced by LiCl (**d**) or Wnt3a (**e**) in 293 T cells. Values are mean ± s.d. of triplicate samples from a representative experiment. AB, siRNA for KDM3A/B; A + B + C, siRNA for KDM3A, KDM3B and JMJD1C. (**f**) KDM3A and KDM3B enhanced Topflash reporter activities induced by β-catenin in 293 T cells. Values are mean ± s.d. for triplicate samples from a representative experiment. β-Cat; the mutant form of β-catenin. (**g**) Expression of KDM3A rescued Wnt3a-induced Topflash activity in 293 T cells depleted of KDM3A/B. Values are mean ± s.d. of triplicate samples from a representative experiment. (**h**) Expression of KDM3A rescued *AXIN2* and *DKK1* expression in 293 T cells depleted of KDM3A/B. Values are mean ± s.d. of triplicate samples from a representative experiment. (**i,j**) Wild-type KDM3A, but not KDM3Amut (KDM3A-H1120A), enhanced Topflash reporter activities induced by β-catenin (**i**) or Wnt3a in 293 T cells. Values are mean and ± s.d. for triplicate samples from a representative experiment. *$P<0.05$; **$P<0.01$, unpaired two-tailed Student' s t-Test. (**k**) Western blot showed the knockdown of KDM3A/B by shRNA in HCT116 cells. (**l**) The knockdown of KDM3A/B inhibited the expression of Wnt target genes by Real-time RT-PCR. (**m**) Quantitative comparison of genes down-regulated by knockdown of KDM3A/B and Wnt/β-catenin target gene signatures in adenoma (left panel) and carcinoma (right panel) using GSEA. $P<0.001$, false discovery rate $q<0.001$.

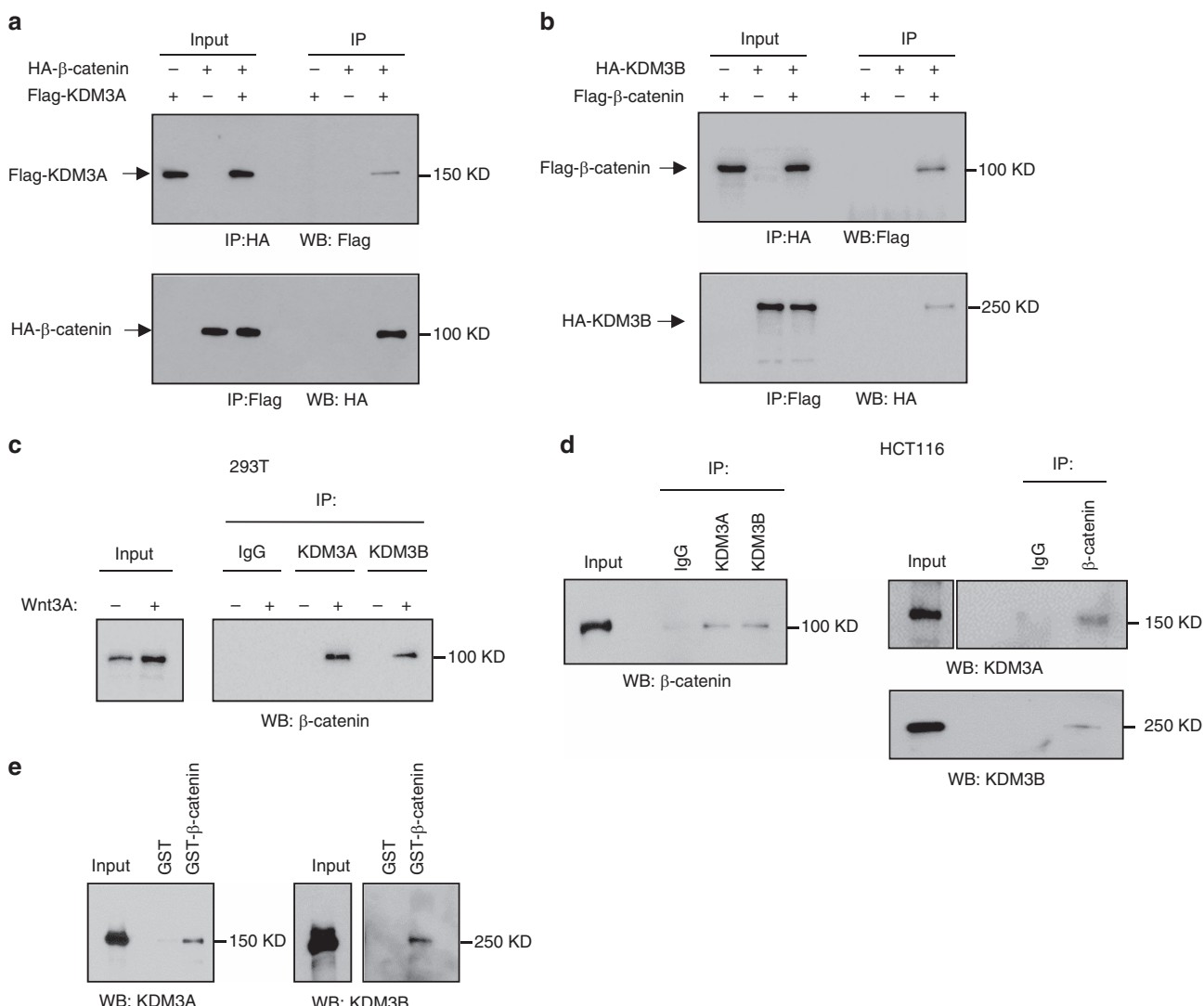

**Figure 2 | KDM3A/B interact with β-catenin.** (**a,b**) KDM3A and KDM3B interacted with β-catenin when overexpressed in 293 T cells. (**c**) Endogenous β-catenin interacted with KDM3A and KDM3B in 293 T cells on Wnt3a stimulation. (**d**) Endogenous β-catenin interacted with KDM3A and KDM3B in HCT116 cells. (**e**) Glutathione S-transferase fusion β-catenin directly interacted with KDM3A and KDM3B in vitro.

studies, we implanted primary human CRC tissues into NOD/SCID mice to generate patient-derived tumour xenografts. We then freshly isolated CSCs from only the first and second generation of CRC xenografts for our mechanistic studies. In addition to ALDH, CD133 and CD44v6 are also utilized as markers to isolate CSCs from human CRCs. Therefore, we utilized ALDH, CD133, and CD44v6 markers to isolate CSCs from the EpCAM$^+$ cell population from xenografts (Fig. 4a, Supplementary Fig. 3a). ALDH$^+$/EpCAM$^+$ cells, CD133$^+$/EpCAM$^+$ and CD44v6$^+$/EpCAM$^+$ effectively formed sphere-like colonies in vitro (Fig. 4b,c, Supplementary Fig. 3a). Notably, the majority of ALDH$^+$ cells were also EpCAM$^+$, which has the same capability to form sphere-like colonies from xenograft tumours (Supplementary Fig. 3b). In addition, a limiting dilution assay showed that ALDH$^+$/EpCAM$^+$ cells had a much greater ability to reform secondary tumours, as compared with ALDH$^-$/EpCAM$^+$ cells (Supplementary Fig. 3c). Knockdown of KDM3A/B by siRNA dramatically inhibited the tumorsphere formation ability of ALDH$^+$/EpCAM$^+$ cells (Fig. 4b–d). Importantly, a significantly higher level of β-catenin/Tcf-mediated transcription was

also observed in ALDH$^+$/EpCAM$^+$ cells compared to ALDH$^-$/EpCAM$^+$ cells (Fig. 4e,f). Moreover, knockdown of KDM3A/B also significantly inhibited the expression of Wnt target genes, including *AXIN2*, *DKK1*, *CCND1* and *MYC* (Fig. 4g). Similarly, knockdown of KDMA/B in CD133$^+$/EpCAM$^+$ and CD44v6$^+$/EpCAM$^+$ also inhibited tumorsphere formation (Supplementary Fig. 3a).

We further examined whether KDM3A/B were required for CSC-mediated tumour formation in vivo. While ALDH$^+$/EpCAM$^+$ cells expressing scramble shRNA (PS1-ALDH$^+$/Scrsh) rapidly formed large tumours in vivo, ALDH$^+$/EpCAM$^+$ cells expressing ABshRNA-1 (PS1-ALDH$^+$/ABsh-1 cells) only formed small tumours or could not form tumour at all (Fig. 4h-j). Similarly, we found that the depletion of KDM3A/B in CSCs isolated from human CRC tissues of the patient in case 2 also inhibited tumorsphere formation in vitro and tumour formation in vivo by inhibiting Wnt/β-catenin signalling (Supplementary Fig. 3d–g).

Previously, CSCs could also be isolated from newly-generated human CRC cell lines HCP1, CC11 and CC12 (ref. 47). Notably, a significantly higher level of β-catenin/TCF-medicated transcription was observed in ALDH$^+$ cells from HCP1 (Fig. 4k)

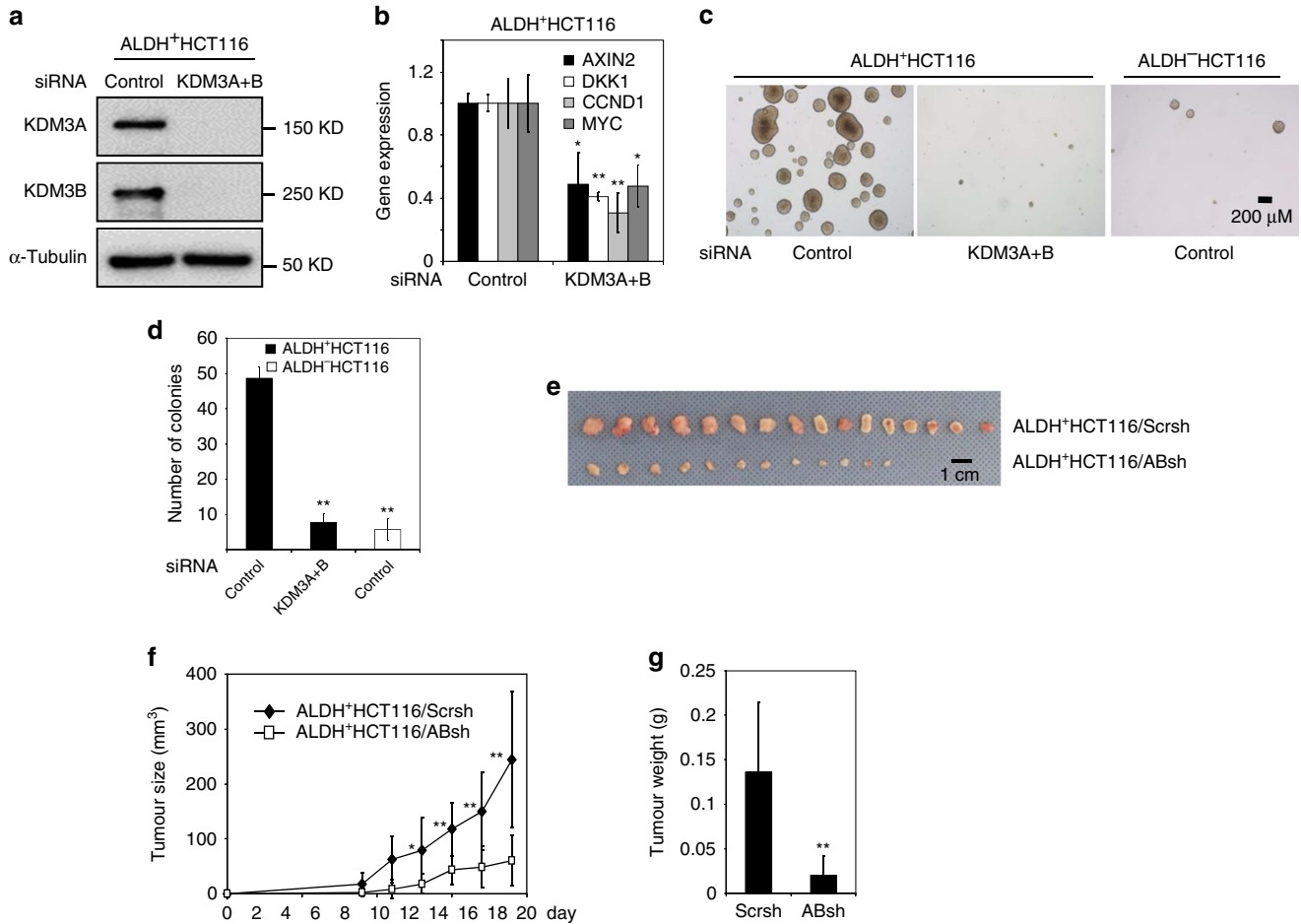

**Figure 3 | KDM3A and KDM3B control the tumorigenic potential of CSC-like cells through Wnt/β-catenin.** (**a**) The knockdown of KDM3A/B by siRNA in ALDH⁺HCT116 cells. (**b**) The knockdown of KDM3A/B inhibited the expression level of *AXIN2, DKK1, CCND1* and *MYC* in ALDH⁺HCT116 cells, *$P < 0.05$, **$P < 0.01$, unpaired two-tailed student's $t$-test ($n = 3$). (**c,d**) The Knockdown of KDM3A/B significantly inhibited tumorsphere formation of ALDH⁺HCT116 cells. The number of spheres were counted from five different fields and averaged. The results represent mean ± s.d. from three independent experiments. *$P < 0.05$, **$P < 0.01$, unpaired two-tailed Student's $t$-test ($n = 3$). (**e,f**) The knockdown of KDM3A/B significantly inhibited tumorigenic potentials of ALDH⁺HCT116 cells *in vivo*. Values are mean ± s.d. from a total of eight mice. *$P < 0.05$, **$P < 0.01$, unpaired two-tailed Student's $t$-test. (**g**) Comparisons of tumour weights at the end of experiments. *$P < 0.05$, **$P < 0.01$, unpaired two-tailed student's $t$-test ($n = 16$).

compared to ALDH⁻ cells. Knockdown of KDM3A/B significantly inhibited the expression of Wnt target genes in ALDH⁺ cells isolated from HCP1 cells (Fig. 4i). Knockdown of KDM3A/B also abolished *in vitro* tumorsphere formation of ALDH⁺ cells from HCP1 cells (Fig. 4m) and from CC11 and CC12 cells (Supplementary Fig. 3h,i). Moreover, knockdown of KDM3A/B also inhibited tumorigenic potentials of ALDH⁺ cells from HCP1 cells in Nude Mice (Fig. 4n).

**KDM3 Promotes CSCs resistance to apoptosis.** Growing evidence suggests that CSCs are resistant to apoptosis induced by chemotherapeutic drugs, which might be responsible for cancer recurrence or relapse[20]. To determine the potential role of KDM3 in chemoresistance of ALDH⁺ cells, we examined whether the depletion of KDM3A/B promoted apoptosis in ALDH⁺HCT116 cells following treatment with two commonly used chemotherapeutic agents cisplatin and irinotecan. Of note, the ALDH⁺ cells were more resistant to cisplatin or irinotecan induced apoptosis than the ALDH⁻ cells (Supplementary Fig. 4a,b). We observed that the transfection of KDM3A/B siRNA alone increased the basal level of cell death in ALDH⁺HCT116 cells. While cisplatin or irinotecan modestly induced cell death in

ALDH⁺HCT116 cells transfected with scramble siRNA, KDM3A/B siRNA significantly enhanced ALDH⁺HCT116 cell death induced by cisplatin and irinotecan (Fig. 5a,b). Western blot analysis revealed that the knockdown of KDM3A/B increased caspase-3 activities induced by cisplatin and irinotecan in ALDH⁺HCT116 cells (Fig. 5c,d). Consistently, knockdown of KDM3A/B also enhanced the cleavage of poly (ADP-ribose) polymerase 1 (PARP1) induced by these two drugs (Fig. 5c,d). Moreover, to rule out non-specific effects of siRNA knockdown, we also overexpressed KDM3B in ALDH⁺HCT116 cells and KDM3A/B knockdown ALDH⁺HCT116 cells (Supplementary Fig. 4c). We found that overexpression of KDM4B promoted ALDH⁺HCT116 cell survival. The restoration of KDM4B expression in KDM3A/B knockdown ALDH⁺HCT116 cells also significantly restored cell resistance to cisplatin and irinotecan (Supplementary Fig. 4d).

To further confirm our results, we also examined whether knockdown of KDM3A/B enhanced apoptosis in ALDH⁺ cells freshly isolated from human CRC tissues. Knockdown of KDM3A/B also significantly enhanced apoptosis induced by cisplatin and irinotecan (Fig. 5e,f) in ALDH⁺ cells from the patient case #1. Western blot also found that the knockdown of KDM3A/B increased caspase-3 activities and cleavage of PARP-1

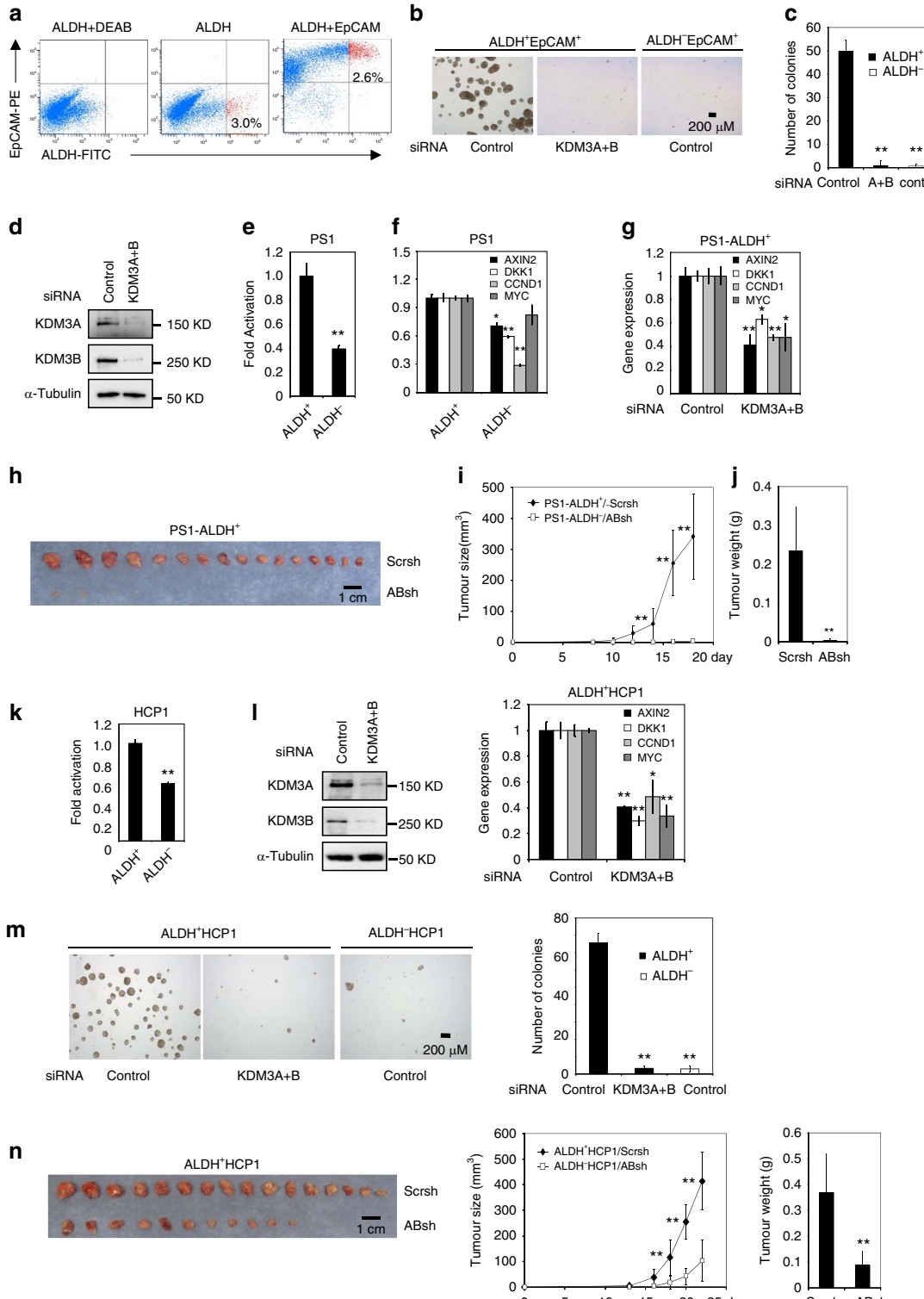

**Figure 4 | KDM3A/B are required to maintain tumorigenic potentials of human colorectal CSCs. (a)** Isolation of ALDH⁺EpCAM⁺ cells from human CRC tissues of patient case #1 (PS1) by FACS. **(b,c)** The knockdown of KDM3A/B inhibited tumorsphere formation of ALDH⁺EpCAM⁺ cells from PS1. **P < 0.01, unpaired two-tailed student's t-test (n = 3). **(d)** The knockdown of KDM3A/B by siRNA in ALDH⁺EpCAM⁺ cells from PS1. **(e)** The Topflash activity was significantly higher in ALDH⁺EpCAM⁺ cells than in ALDH⁻EpCAM⁺ cells from PS1. **(f)** The expression of Wnt target genes was higher in ALDH⁺EpCAM⁺ cells than in ALDH⁻EpCAM⁺. **(g)** The knockdown of KDM3A/B inhibited the expression of AXIN2, DKK1, CCND1 and MYC in ALDH⁺EpCAM⁺ cells from PS1 **(h–j)** The knockdown of KDM3A/B significantly inhibited tumour growth of ALDH⁺EpCAM⁺ cells in Nude mice. Tumour growth **(i)** was monitored for 3 weeks and the weight of the tumors from each group **(j)** were compared at the end of experiments. **P < 0.01, unpaired two-tailed student's t-test (n = 16). **(k)** The Topflash activity was significantly higher in ALDH⁺HCP1 cells than in ALDH⁻HCP1 cells. **(l)** The knockdown of KDM3A/B in ALDH⁺HCP1 cells inhibited the expression of *AXIN2, DKK1, CCND1* and *MYC*. **(m)** The knockdown of KDM3A/B inhibited tumorsphere formation of ALDH⁺HCP1 *in vitro*. **(n)** The knockdown of KDM3A/B inhibited the tumorigenic potential of ALDH⁺HCP1 cells *in vivo*. **P < 0.01, unpaired 2-tailed Student's t-test (n = 16).

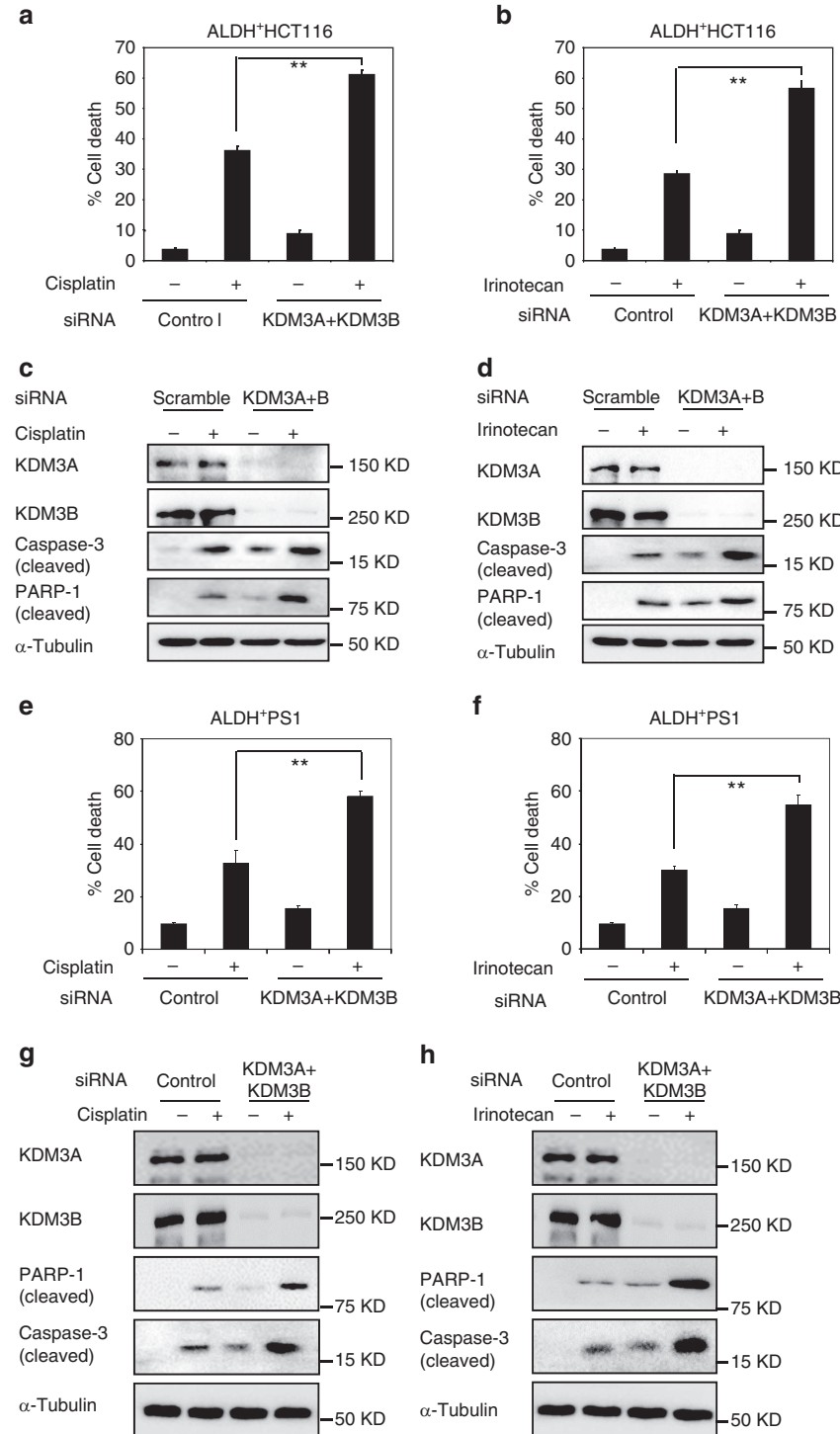

**Figure 5 | KDM3A/B promote chemoresistance in human CSCs.** (**a**) The knockdown of KDM3A/B enhanced cell death induced by cisplatin in ALDH⁺HCT116 cells. Values are mean ± s.d. of triplicate samples from a representative experiment. *$P<0.05$, **$P<0.01$, unpaired two-tailed Student's t-test ($n=3$). (**b**) The knockdown of KDM3A/B enhanced cell death induced by irinotecan in ALDH⁺HCT116 cells. Values are mean ± s.d. of triplicate samples from a representative experiment. *$P<0.05$, **$P<0.01$, unpaired two-tailed Student's t-test ($n=3$). (**c**) The knockdown of KDM3A/B enhanced the capase-3 activation and the cleavage of PARP-1 in ALDH⁺HCT116 induced by cisplatin. (**d**) The knockdown of KDM3A/B enhanced the capase-3 activation and the cleavage of PARP-1 in ALDH⁺HCT116 induced by irinotecan. (**e**) The knockdown of KDM3A/B enhanced cell death induced by cisplatin in ALDH⁺ cells from PS1. **$P<0.01$, unpaired two-tailed Student's t-test ($n=3$). (**f**) The knockdown of KDM3A/B enhanced cell death induced by irinotecan in ALDH⁺ cells from PS1. **$P<0.01$, unpaired 2-tailed Student's t-Test ($n=3$). (**g**) The knockdown of KDM3A/B enhanced the capase-3 activation and the cleavage of PARP-1 in ALDH⁺ cells from PS1 induced by cisplatin. (**h**) The knockdown of KDM3A/B enhanced the capase-3 activation and the cleavage of PARP-1 in ALDH⁺ cells from PS1 induced by irinotecan.

induced by cisplatin or irinotecan in ALDH$^+$ CSCs from patient case #1 (Fig. 5g,h).

**KDM3 directly binds to Wnt target gene promoters.** Our co-IP assays found that β-catenin interacted with KDM3A/B. KDM3A/B have a preference for H3K9me2, and they exhibit weak or no activity toward H3K9me3 (refs 36,37). To determine how KDM3A/B globally controlled gene expression by erasing H3K9me2, we performed ChIP-seq to determine whether knockdown of KDM3A/B affected the levels of H3K9me2 on Wnt target gene promoters. Since knockdown of KDM3A/B impaired CSC growth, we first performed ChIP-seq in HCT116 cells because we needed a large amount of cells for ChIP-seq. Due to the fact that ChIP-seq signals obtained from anti-KDM3A antibodies had strong background noise, we focused our analysis on KDM3B. Globally, the majority of KDM3B peaks were found within ± 5 kb away from transcription start site (TSS), especially within ± 2 kb away from the TSS (Fig. 6a,b). In addition, the cis-regulatory element annotation system (CEAS) software revealed that KDM3B was enriched at the promoter regions (16.5% for KDM3B ChIP and 1.1% for Input) (Supplementary Fig. 5a). Moreover, we found that chromosomes 1, 17, 19, 22 were significantly ($P < 10^{-34}$) enriched for KDM3B binding (Supplementary Fig. 5b). These data suggest that KDM3B binding to the genome was a non-random occurrence, with preferential binding to certain chromosomes and promoter regions. Globally, the percentage of occupation of H3K9me2 peaks ± 5 kb away from the TSS was only 5.8% (Fig. 6a). Notably, the KDM3B enrichment within ± 5 kb of TSS site decreased in KDM3A/B-depleted cells, corresponding to a dramatic increase in H3K9me2 enrichment at the same region as compared to control cells (Fig. 6b). CEAS software revealed that H3K9me2 was enriched not only at the promoter regions but also enriched at the whole gene body region including intron, coding region and 3′-UTR following KDM3A/B depletion (Supplementary Fig. 5a). Of note, gene ontology analysis showed that a remarkable number of the genes with enhanced H3K9me2 signals were associated with the Wnt/β-catenin signalling pathway (Fig. 6c).

Next, we investigated whether KDM3B could specifically bind to the Wnt target gene promoters and control H3K9me2 patterns. We grouped the TSS of 31 Wnt target genes and centred all regions. These Wnt targets were selected from the Wnt homepage and had been reported to be directly upregulated by β-catenin (Supplementary Table 2). We then plotted the average levels of KDM3B peaks within ± 5 kb of each Wnt target TSS region center. As shown in Fig. 6d, the average profile of KDM3B enrichment decreased corresponding to a dramatic increase in H3K9me2 enrichment following KDM3A/B depletion as compared to control cells. ChIP-seq demonstrated that KDM3B bound to the well-known Wnt target genes, including AXIN2, DKK1, CCND1 and MYC. The knockdown of KDM3A/B significantly increased the levels of H3K9me2 on these genes (Fig. 6e–h).

We further performed individual ChIP-qPCR to examine the enrichment of KDM3B and H3K9me2 on AXIN2, DKK1, CCND1 and MYC. KDM3A/B depletion resulted in enhanced H3K9me2 enrichment in both intron and coding exon regions of AXIN2, which was validated by qPCR targeted at the Wnt-responsive-element (WRE), introns and coding exons (A1–A5). As a strict negative control (NEG), a region located in the 10 kb downstream of the TTS was also examined (Supplementary Fig. 6a). Furthermore, the recruitment of β-catenin to these regions, especially at the WRE, was not affected following KDM3A/B depletion (Supplementary Fig. 6a). Similar findings were observed at the WRE, intron and distal intergenic region of

DKK1 (Supplementary Fig. 6b), CCND1 (Supplementary Fig. 6c) and MYC (Supplementary Fig. 6d).

To validate that KDM3B could specifically bind to the Wnt target gene promoters and regulate H3K9me2 patterns, HA-tagged KDM3B was expressed in HCT116 cells. ChIP-qPCR demonstrated that the recruitment of both KDM3B and HA to AXIN2 WRE was significantly enhanced in parallel with decreased H3K9me2 enrichment (Supplementary Fig. 6e,f). Moreover, the recruitment of β-catenin to WRE was not affected by the expression of KDM3B (Supplementary Fig. 6e). Similar results were also detected at the WRE region of DKK1 and CCND1 (Supplementary Fig. 6g,h). Our ChIP-qPCR results also revealed that KDM3A was present on these Wnt target gene promoters (Supplementary Fig. 7a,b). Similar results were also obtained in SW480 cells (Supplementary Fig. 7c–f).

**KDM3 coordinates H3K9/H3K4 Methylation.** On the basis of our ChIP-seq results, we also examined the occupancy of KDM3A/B, as well as H3K9me2 marks on the Wnt target promoters in ALDH$^+$ cells isolated from HCP-1 cells in which Wnt/β-catenin signalling was constitutively activated. ChIP-qPCR showed that KDM3A/B were present on the AXIN2 promoter in ALDH$^+$ cells isolated from HCP1 cells (Fig. 7a). KDM3A/B depletion resulted in increased H3K9me2 enrichments on the AXIN2 promoter while they did not affect β-catenin binding to their promoters (Fig. 7b,c). Similarly, KDM3A/B were detected on the DKK1 promoter, and the depletion of KDM3A/B elevated H3K9me2 marks on the DKK1 promoter without interfering with β-catenin occupancy on the chromatin (Fig. 7d–f). Since Wnt/β-catenin-mediated transcription was also hyper-activated in ALDH$^+$ cells isolated from human CRC xenografts, we further confirmed our results in these cells. We found that KDM3A/B were enriched on the promoters of AXIN2 and DKK1 in ALDH$^+$ cells isolated from human CRC xenografts. KDM3A/B depletion also increased H3K9me2 marks on the promoters of AXIN2 and DKK1, but did not affect β-catenin levels on chromatin (Supplementary Fig. 8a–f).

Gene activation often requires the coordinated removal of silencing marks by histone demethylases and the addition of activating marks by methyltransferases. H3K4me3 is a hallmark for gene activation and it is critical for recruiting the transcriptional co-activator BCL9/PYGO2 to activate Wnt target gene transcription. Unexpectedly, following KDM3A/B depletion, the enrichment of H3K4me3 at the WRE of AXIN2 was significantly reduced in ALDH$^+$ cells isolated from HCP1 cells (Fig. 7g). Since H3K4me3 is methylated by the H3K4-specific methltransferase mixed lineage leukaemia 1 (MLL1) or MLL2, we examined whether KDM3AB depletion affected MLL1 and MLL2 binding to the Wnt target gene promoters. We found that MLL1, but not MLL2, was present on the AXIN2 promoter in ALDH$^+$ cells. KDM3A/B knockdown significantly decreased recruitment of MLL1 to the AXIN2 promoter (Fig. 7h). Similarly, ChIP-qPCR also revealed that KDM3A/B knockdown impaired MLL1 binding to the DKK1 promoter and reduced H3K4me3 enrichment in ALDH$^+$ cells (Fig. 7i,j). Our results suggest that KDM3A/B might act in concert with histone methyltransferases to orchestrate Wnt target gene activation. Furthermore, co-IP assays showed that MLL1, but not control IgG, was able to pull down Flag-tagged KDM3A or HA-tagged KDM3B in 293 T cells (Supplementary Fig. 8g). Moreover, Co-IP assays also revealed that endogenous MLL1 proteins interacted with endogenous KDM3A and KDM3B in HCT116 cells (Supplementary Fig. 8h). On Wnt stimulation, β-catenin recruits other transcriptional co-activators such as BCL9 and PYGO2 to Wnt target gene

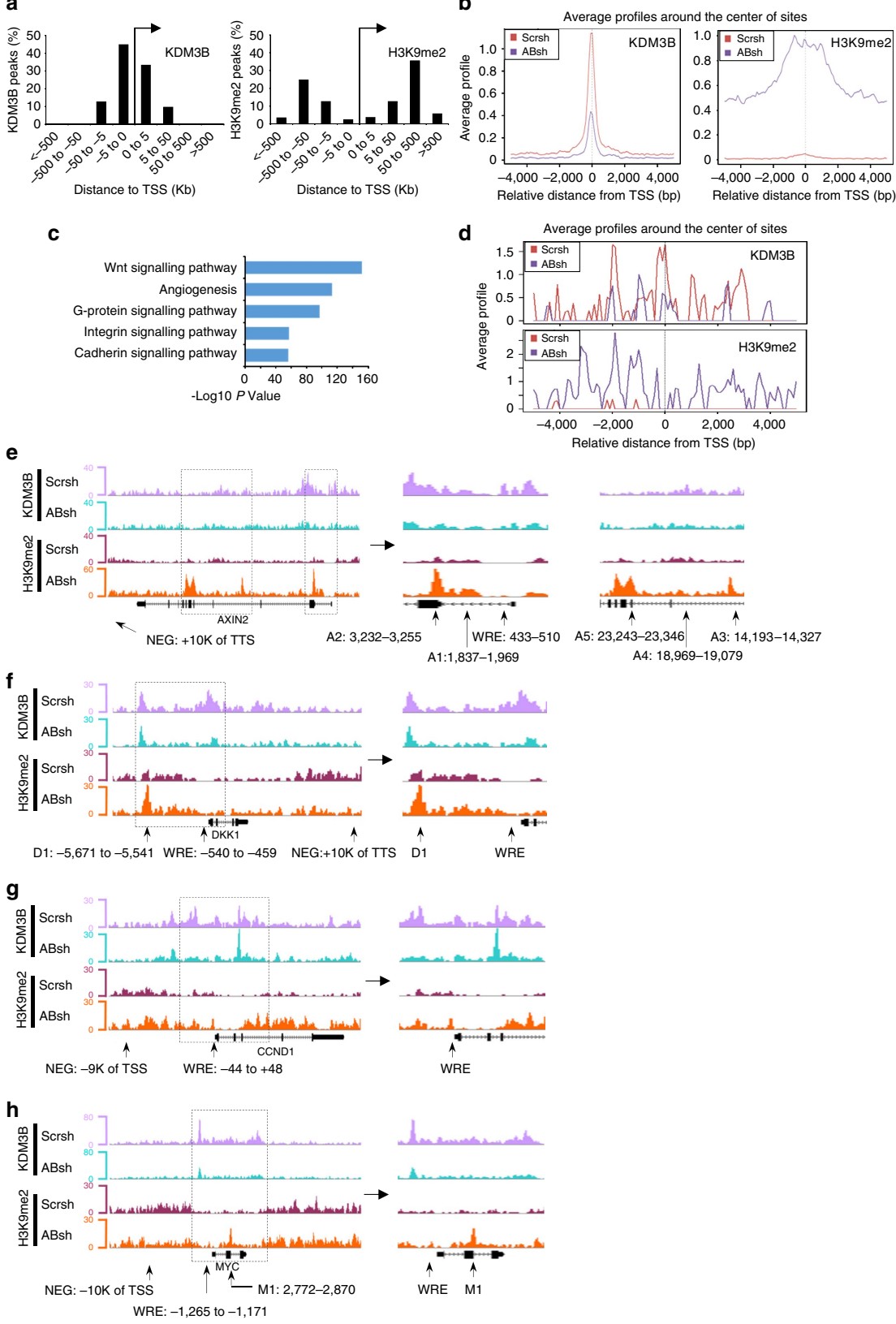

**Figure 6 | KDM3A/B bind to Wnt target gene promoters to erase H3K9me2 marks.** (**a**) The bar chat shows the distribution of distances between KDM3B or H3K9me2 peaks and TSS. (**b**) Average levels of KDM3B and H3K9me2 enrichment within ± 5 kb of TSS in HCT116/ABsh-1 cells were compared to HCT116/Scrsh cells. (**c**) Gene ontology analysis of enriched H3K9me2 target genes. The x-axis value is − $\log_{10}$ of binomial raw P value. (**d**) Average levels of KDM3B and H3K9me2 around the Wnt target genes transcription start site in HCT116/Scrsh cells were compared to HCT116/ABsh-1 cells. (**e–h**) Representative view of ChIP-seq results in *AXIN2*, *DKK1*, *CCND1* and *MYC* gene by Genome Browser.

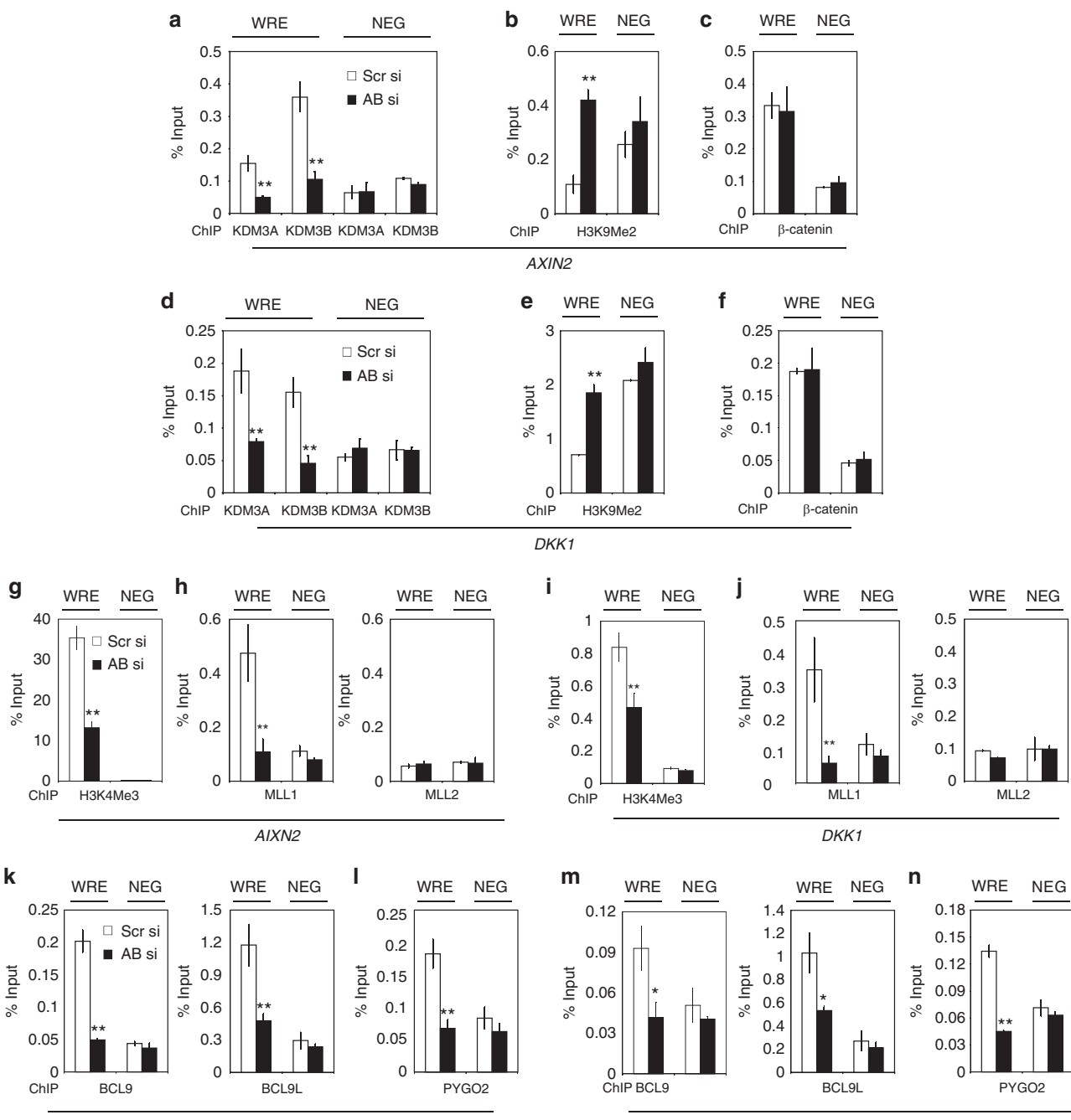

**Figure 7 | KDM3A/B directly erase H3K9 methylation and promote H3K4 methylation by MLL1.** (**a**) ChIP assays confirmed that the knockdown of KDM3A/B abolished KDM3A/B enrichments on the *AXIN2* promoter in ALDH+HCP1 cells. (**b**) ChIP assays confirmed that the knockdown of KDM3A/B increased the levels of H3K9me2 on the *AXIN2* promoter in ALDH+HCP1 cells. (**c**) ChIP assays showed that the knockdown of KDM3A/B did not affect β-catenin binding on the *AXIN2* promoter in ALDH+HCP1 cells. (**d**) ChIP assays confirmed that the knockdown of KDM3A/B abolished KDM3A/B enrichments on the *DKK1* promoter in ALDH+HCP1 cells. (**e**) ChIP assays confirmed that the knockdown of KDM3A/B increased the levels of H3K9me2 on the *DKK1* promoter in ALDH+HCP1 cells. *P<0.05, **P<0.01, unpaired two-tailed Student's *t*-test (*n* = 3). (**f**) ChIP assays showed that the knockdown of KDM3A/B did not affect β-catenin binding on the *DKK1* promoter in ALDH+HCP1 cells. *P<0.05, **P<0.01, unpaired 2-tailed Student's *t*-test (*n* = 3). (**g**) ChIP assays showed that the knockdown of KDM3A/B reduced the levels of H3K4me3 on the *AXIN2* promoter in CSCs isolated HCP1 cells. (**h**) ChIP assays showed that the knockdown of KDM3A/B reduced the recruitment of MLL1 to the *AXIN2* promoter in ALDH+HCP1 cells. (**i**) ChIP assays showed that the knockdown of KDM3A/B reduced the levels of H3K4me3 on the *DKK1* promoter in ALDH+HCP1 cells. (**j**) ChIP assays showed that the knockdown of KDM3A/B reduced the recruitment of MLL1 to the *DKK1* promoter in ALDH+HCP1 cells. (**k**) ChIP assays showed that KDM3A/B knockdown reduced BCL9/BCL9L occupancies on the *AXIN2* promoter in ALDH+HCP1 cells. (**l**) ChIP assays showed that KDM3A/B knockdown reduced PYGO2 occupancies on the *AXIN2* promoter in ALDH+HCP1 cells. (**m**) ChIP assays showed that KDM3A/B knockdown reduced BCL9/BCL9L occupancies on the *DKK1* promoter in ALDH+HCP1 cells. (**n**) ChIP assays showed that KDM3A/B knockdown reduced PYGO2 occupancies on the *DKK1* promoter in ALDH+HCP1 cells.

promoters[11–15]. PYGO2 binds to H3K4me3 and H3K4me2 on Wnt target gene promoters to promote transcription in colorectal cancer cells[32,33]. Therefore, we examined whether knockdown of KDM3A/B affected the presence of PYGO2 and BCL9 and its closely related homologue BCL9L on the promoters of *AXIN2* and *DKK1* in CSCs with constitutively active Wnt/β-catenin signalling. ChIP-qPCR revealed that knockdown of KDM3A/B significantly inhibited BCL9/BCL9L, as well as PYGO2 binding to the *AXIN2* promoter in ALDH[+] cells (Fig. 7k,l). Similarly, the knockdown of KDM3A/B also impaired the recruitment of BCL9/BCL9L and PYGO2 to the DKK1 promoter in ALDH[+] cells (Fig. 7m,n).

## Disscussion
In this study, we identified that KDM3 family histone demethylases play a critical role in β-catenin/Tcf-mediated transcription by demethylating H3K9me2. Mechanistically, we showed that KDM3A/B not only demethylated H3K9me2 but also indirectly promoted H3K4 methylation by recruiting MLL1. In consequence, methylated H3K4 promoted the recruitment of BCL9/PYGO to the chromatin, promoting Wnt target gene transcription. Our results provide novel insights into how repressive and active epigenetic histone marks are epigenetically modified and subsequently how they regulate β-catenin/Tcf-mediated transcription. Importantly, we showed that KDM3 plays a critical role in tumorigenic potentials and survival of CSCs from human CRCs, highlighting that KDM3 might be a new therapeutic target for effective elimination of human colorectal CSCs.

The KDM3 family proteins are H3K9me2/me1 demethylases with a preference for dimethylated residues. Most studies have showed that they promote gene transcription by erasing H3K9me2 (refs 36,37,48). Consistently, we found that KDM3A/B weakly affected H3K9me1 (Supplementary Fig. 6a). H3K9 methylation is generally involved in transcriptional repression and heterochromatin formation[49,50]. Multiple studies have demonstrated that H3K9me2 and H3K9me3 are enriched in the transcriptional start sites of silenced genes[51–53]. These findings highlight that the methylation state of H3K9 is an important mark for the transcriptional status of a given gene. Our results suggested that both KDM3A and KDM3B were recruited to Wnt target gene promoters, erasing H3K9me2 to activate Wnt target gene transcription. On the other hand, histone H3K4 di- and tri-methylation (H3K4me2/me3) are associated with active gene transcription[54–56]. Both H3K4 and H3K9 methylation states are regulated by specific series of demethylases and methyltransferases[57,58]. Balance between H3K4 and H3K9 methylation has been demonstrated as a critical determinant of the boundaries between euchromatin and heterochromatin[49,50]. Histone demethylases could be associated with other chromatin remodelers to regulate gene transcription[59–61]. Our studies found a novel coordination between histone demethylation of H3K9 by KDM3 and methylation of H3K4 by MLL1 to activate target gene expression upon Wnt activation. As KDM3 and MLL1 exist in a complex recruited by β-catenin, KDM3 could also play an important role in the regulation of H3K9 and H3K4 methylation balance in Wnt signalling. Interestingly, KDM3B has been reported to interact with histone acetyltransferase CBP and promote leukemogenesis via activation of *lmo2* (ref. 37). Considering that CBP is also an important transcriptional co-activator in Wnt signalling, we speculate that KDM3 may also interact with CBP and regulate the balance between H3K9 demethylation and H3/H4 acetylation at Wnt target gene promoters.

H3K4me3 is critical for recruiting PYGO to chromatin[31–33]. BCL9 and PYGO are important transcriptional co-activators which promote β-catenin/Tcf-mediated transcription in colorectal cancer cells[11,12,62–64]. BCL9 is an adaptor between β-catenin and PYGO[11],

and it assists PYGO in recognizing modified histone H3 tails (H3K4me3/me2) via their plant homeodomain fingers[32]. While it is well-known that H3K4 methylation is critical for Wnt gene transcription, until now, it has been unclear whether removing H3K9 methylation is required for activating β-catenin/Tcf-mediated transcription. Our ChIP-seq analysis revealed that H3K9me2 in the promoter of Wnt target genes is highly enriched upon KDM3A/B depletion. In other words, it is critical to erase H3K9me2 in order to activate the transcription of Wnt target genes. Importantly, we found that KDM3A/B also interacted with MLL1 and promoted MLL1 binding to the Wnt target gene promoter. The depletion of KDM3A/B impaired MLL1 binding to the Wnt target gene promoters. Therefore, it is possible that β-catenin and KDM3A/B might coordinate to recruit MLL1 to the Wnt target gene promoters, thereby enhancing H3K4 methylation and promoting PYGO/BCL9 to the Wnt target gene promoters to activate transcription.

The CSC model has been implicated in tumour therapy resistance and tumour recurrence[20]. To understand the epigenetic mechanisms that regulate the CSC tumorigenic potential will help to develop new targeting strategies to eliminate the CSC population and improve the clinical outcomes of patients with CRC. Two different models of CRC initiation associated with CSCs have been proposed[65]. On the basis of genetic studies, it has been proposed that Lgr5[+] intestinal stem cells from the crypt bottom are readily targeted by Wnt signalling and play a key role in CRC initiation[66,67]. Another model involves the dedifferentiation process in which the cells re-acquire CSC-like properties[68,69]. Elevated NF-κB signalling has been shown to enhance Wnt activation and induce dedifferentiation of non-CSCs in the colon that acquired tumour-initiating capacity[68]. Interestingly, we found that knockdown of KDM3A/B inhibited the tumorigenic potential and self-renewal of CSCs. KDM3A/B have been found to be upregulated in human CRC tissues. It is possible that they might help non-stem tumour cells to epigenetically reacquire CSC-like properties by promoting Wnt/β-catenin signalling in human CRCs. Another important property of CSCs is therapy resistance. Notably, we found that the depletion of KDM3A/B promoted CSC apoptosis and sensitized CSC to chemotherapeutic drug-mediated apoptosis. Our results suggest that targeting KDM3 might help to eliminate not only non-stem tumour cells, but also tumour propagating CSCs in human CRCs by inhibiting Wnt/β-catenin signalling.

## Methods
**Plasmidsand antibodies.** Flag-KDM3A plasmids were kindly provided by Dr Yi Zhang. Flag-KDM3A-H514A and HA-KDM3A constructs were prepared using PCDNA3.1+ by standard site-directed mutagenesis and sub-cloning. The pOTB7-KDM3B was purchased from GE Healthcare and subcloned into PCDNA3.1+ expression vector to generate HA-KDM3B. The Topflash reporter and CMV-β-galactosidase were used as previously described[40]. The primary antibodies were purchased from the following resources: anti-β-catenin (BD, #610153; 1:10,000); anti-Flag (Sigma-Aldrich, F1804 and F7425; 1:10,000); anti-α-tubulin (Sigma-Aldrich, T6074; 1:10,000); anti-HA (Covance, MMS-101P; 1:5,000); anti-TBP (Abcam, ab818; 1:2,000); anti-BCL9 (Abcam, ab37305; 1:1,000); anti-H3K9me2 (Abcam, ab1220; 1:500); anti-H3K4Me3 (Abcam, ab8580; 1:500); anti-KDM3A (Bethyl Laboratory, A301-539A: 1:5,000); anti-PYGO2 (Bethyl Laboratories, A310-999A; 1:5,000); anti-BCL9L (Bethyl Laboratories, A303-152A; 1:10,000); anti-MLL1 (Bethyl Laboratories, A300-374A; 1:5,000); anti-MLL2 (Bethyl Laboratories, A302-175A; 1:5,000); anti-KDM3B (Millipore, 09-816; 1:5,000) and anti-JMJD1C (Millipore, 09-817; 1:5,000); anti-PARP1 (cleaved form) (Cell Signaling, #5625; 1:2,000) and anti-Caspase-3 (cleaved form) (Cell Signaling, #9661; 1:2,000). All histone demethylase-targeted siRNA and scramble siRNA were purchased from Santa Cruz Biotechnology. Each siRNA consisted of pools with three to five target-specific 19–25 nt siRNAs designed to knockdown target gene expression. The siRNA sequence targeting 3′-UTR of KDM3A is 5′-GAAATCA ACTACTGTACAA-3′. The shRNA sequences for the knockdown of KDM3A and KDM3B simultaneously are 5′-GATCTGGGCCCCAAGATGTA-3′ (KDM3AB-1) and 5′-AACAAATCTTCACTTAGATGT-3′ (KDM3AB-2). The shRNA sequences for the knockdown of JMJD1C is 5′-GCGGAATCAATTAGTCTTGAT-3′.

All other sequences of RT-PCR and ChIP primers are listed in Supplementary data (Supplementary Table 3) or as previously described[10].

**Cell culture and siRNA screening.** 293 T, HCT116 and SW480 cells were purchased from ATCC and maintained in DMEM medium containing 10% fetal bovine serum (FBS) and antibiotics (penicillin and streptomycin). Cell lines have been authenticated by polymorphic short tandem repeat (polymorphic short tandem polymorphic short tandem repeat polymorphic short tandem repeat STR) profiling. The newly-generated human CRC cell lines HCP-1, CC11 and CC12 were from the Dr Ellis Lee Lab and were cultured in DMEM/F12 supplemented with 10% FBS, penicillin-streptomycin, sodium pyruvate, L-glutamine, and nonessential amino acids at 37 °C in 5% $CO_2$.

Lentiviral vector pGreenFir1-TCF/LEF1 and packaging pPACK-H1 Lentiviral Packaging Plasmid Mix were products of System Bioscience. For viral transduction, lentivirus was obtained by co-transfection of pGreenFir1-TCF/LEF1 with packaging plasmids into 293 T cells. To generate 293 T/Top stable cell lines, 293 T cell were seeded in 10 cm dishes and incubated overnight. The next day cells were treated with viral particles. After 12 h, the medium was replaced with fresh medium. Four days post virus infection, 20 mM LiCl was added in to the medium with incubation for 10 h. The cells were trypsinized to form single cells and GFP+ cells were separated by fluorescence-activated cell sorting using a FACSAria III (BD Immunocytometry Systems). After sorting, the GFP+ cells were grown for at least three passages for further studies. 293 T/Top stable cells were transfected with siRNAs targeting histone demethylases using Lipofectamine RNAiMax (Invitrogen). For luciferase assays, 36 h post transfection, 20 mM LiCl was added in to the medium followed by 12 h incubation. Luciferase activity of total cell lysates was measured using Luc-Screen kits (Tropix). The Topflash activity was normalized against the protein concentration of each cell lysate sample. For measurement of knockdown efficiency, 48 h after transfection, total RNA was isolated from cells from the same batch of experiments using TRIzol reagents, and cDNA was synthesized with oligo(dT) primers using M-MuLV reverse transcriptase (NEB). Quantitative RT-PCR (qRT-PCR) analysis was carried out with iQ SYBR green supermix (Bio-Rad) on an iCycler iQ real-time PCR detection system (Bio-Rad).

For Topflash reporter assays, 293 T cells or CRC cells were plated at 40–50% confluency. Approximately 100 ng of Topflash and 50 ng of CMV-β-galactosidase constructs and other expression vectors were transfected using Lipofactamine 2000 (Invitrogen). For siRNA combined with luciferase reporter assays, DNA constructs were transfected 24 h post siRNA transfection. To knock down KDM3A and KDM3B by shRNA, AB-1, AB-2 shRNA or control shRNA were cloned into with pLKO.1-puro vector (Addgene).

**Western blot and immunoprecipitation.** For immunoprecipitation (IP), 293 T cells or human colorectal cancer cells ($1 \times 10^6$) were lysed in 500 ml of CelLytic M buffer (Sigma) with a protease inhibitor cocktail (Roche) for 10 min on ice. After centrifugation at $10,000g$ at 4 °C, the supernatant was transferred into a new tube and incubated with primary antibodies at 4 °C for 2 h, followed by incubation with a 20 μl bed volume of protein A or protein G-Sepharose (GE Healthcare) for an additional 1 h. Immunoprecipitates were washed three times with PBS with 0.1% NP-40 buffer at 4 °C. Proteins bound to the beads were eluted with SDS-loading buffer at 98 °C for 2 min, subjected to SDS–polyacrylamide gel electrophoresis (SDS–PAGE) and then transferred to the a polyvinylidene difluoride membrane, followed by western blotting analysis using an ECL plus kit (Millipore). The important original immunoblotting are shown in Supplementary Fig. 9.

**Microarray and gene set enrichment analysis.** For microarray, total RNA was extracted using RNeasy kit (Qiagen). The biotin-labelled RNAs were fragmented and hybridized with an Affymetrix Human Genome U133 Plus 2.0 Array at the UCLA DNA Microarray Core Facility. The arrays were scanned with the GeneArray scanner (Affymetrix). The robust multichip average method was used to normalize the raw data. GSEA were performed with GSEA software (http://www.broad.mit.edu/GSEA). This is a computational method that determines whether an a priori defined set of genes shows statistically significant, concordant differences between two biological states (for example, phenotypes). P values were computed using a bootstrap distribution created by resampling gene sets of the same cardinality. Lists of Wnt targets in Adenoma ($n = 157$) and Carcinoma ($n = 148$) were described previously[38].

**Isolation of CSCs.** The discarded human CRC tissues were provided by the UCLA Translational Pathological Core and were approved by the UCLA Institutional Review Board. The tumour tissues were cut into small pieces around 2 mm in diameter and implanted into the flanks of NOD-SCID mice. About 6–8 weeks after implantation the xenograft tumours were isolated from mice, minced and dissociated with collagenase IV (1 mg ml$^{-1}$; Worthington Biochemical). The resulting single-cell suspension was passed through a 70-μm mesh filter followed by a 40-μm mesh filter. Red blood cells from the surgical samples were lysed using a Cell Lysis Buffer (Sigma). Flow cytometry used EpCAM as a marker to obtain carcinoma cell-enriched populations by eliminating stromal cells[70]. For CD133+ CSCs isolation, APC-coupled anti-CD133 (Miltenyi Biotec), and PE-coupled anti-EpCAM antibodies (Miltenyi Biotec) were used (1:20 dilution) and CD133+ EpCAM+ CSCs were sorted using a fluorescence-activated cell sorter (FACSAria III, BD Immunocytometry Systems). Non-specific binding was blocked by FBS (2%; Invitrogen), and controls included IgG alone. For identification of ALDH+ CSCs, cells from human CRC tissues were incubated with PE-coupled anti-EpCAM antibodies and subsequently stained with anti-ALDEFLUOR kits (Stem Cell Technologies). In all experiments, the ALDEFLUOR-stained cells were treated with diethylaminobenzaldehyde, a specific ALDH inhibitor that served as ALDH-negative controls. For ALDH+ or CD133+ cells from human colorectal cell lines including HCT116, SW480, HCP1, CC11 and CC12, cells were not stained with PE-coupled anti-EpCAM antibodies. ALDH+ or CD133+ subpopulations were separated from these cells by a fluorescence-activated cell sorting.

**Mouse xenografts and tumoursphere formation assays.** For tumoursphere formation assay, the sorted single cells were plated on ultralow attachment six-well plates (Corning) at a density of 25,000 viable cells per well for HCT116 and SW480 cells. Freshly isolated CRC cells (HCP1, CC11 and CC12) and for cells from xenografts (patient case 1 and 2) were plated at 10,000 viable cells per well. Cells were grown in a serum-free mammary epithelial basal medium (MEBM; Lonza), supplemented with B27 (Invitrogen), 20 ng ml$^{-1}$ EGF (R&D Systems), 10 ng ml$^{-1}$ FGF (R&D Systems), 4 mg ml$^{-1}$ Gentamycin (Invitrogen), 1 ng ml$^{-1}$ Hydro-cortisone (Sigma-Aldrich), 5 mg ml$^{-1}$ Insulin and 100 mmol l$^{-1}$ beta-mercap-toethanol (Sigma-Aldrich). For siRNA transfection, 1 h after plating, the cells were transfected with KDM3A/B siRNA or control siRNA. Tumourspheres were observed under the microscope 2 weeks later. For shRNA knockdown, CSCs were infected with the lentiviruses expressing Scr shRNA or ABsh-1 for 24 h. After rapid selection with puromycin, $5 \times 10^3$ cells were mixed with an equal volume of Matrigel and injected subcutaneously into the flank area of 6–8-week-old female nude mice. We used a sample size of 8 mice per group to reach statistical significance based on our previous experiences. Tumour growth was determined and measured as previously described[10]. We housed mice in pathogen-free facilities under 12-h light and 12-h dark cycle. The animal protocol and experimental procedures were approved by the Division of Laboratory Animal Medicine of UCLA and were in accordance with the US National Institute of Health guidelines. The animals were randomly assigned to procedure groups. However, not all animal experiments were conducted in a completely blinded fashion.

**Immunostaining and statistical analysis.** TMAs were purchased from US Biomax. Colon Cancer Screen TMA (BC05118a) contains 50 colon adenocarcinoma specimens with matching adjacent normal tissues. For immunostaining, tissue antigens were retrieved as described previously[40]. The slides were then stained with monoclonal antibodies against β-catenin (1:100; BD Biosciences), polyclonal antibodies against KDM3A (1:100; Bethyl), and polyclonal antibody against KDM3B (1:200; Millipore) at 4 °C overnight in a humid chamber (Sigma). We then incubated the sections with HRP-labelled polymer for 60 min at room temperature, detected the immunocomplexes with AEC+ chromogen (Dako EnVision System) and counterstained with hematoxylin QS (Sigma). The intensity of immunostaining was scored as follows: 0, no staining; +, weak staining; ++, moderate staining; +++, strong staining. The Wilcoxon signed rank test was used to explore the significant differences in IHC staining intensity between different groups. All statistical analyses were performed using the SPSS 17.0 software.

**ChIP-qPCR and ChIP-seq Analysis.** For ChIP-qPCR assays, cells were treated for 10 min with 5 mM dimethyl 3,3′-dithiobispropionimidate-HCl (DTBP) (Pierce) in PBS at room temperature, rinsed with 100 mM Tris–HCl, 150 mM NaCl (pH 8.0) and crosslinked with 1% formaldehyde in PBS at 37 °C for 10 min. Total cell lysates were sonicated to generated 200–500 bp DNA fragments. IP was performed as described previously[10]. All resulting precipitated DNA samples were quantified by quantitative real-time PCR (qPCR). Data are expressed as the percentage of input DNA. The primer sequences used for qPCR were as listed in Table 6.

ChIP-seq libraries were prepared according to the NuGEN protocol and sequenced using an Illumina Genome Analyzer. All sequencing reads (50 bp in length) were mapped to NCBI build 37 (hg19) of the human genome using the Bowtie software. The mapped reads were subjected to the algorithm to evaluate the bound regions (peaks) of these reads in the genome. In detail, the genome was divided into 100-bp windows and we calculated the P value for Poisson distribution of ChIP-ed DNA relative to input for each window. The statistical cutoff for significant peaks was $P < 10^{-3}$ and with two neighbouring windows of the same significance. Here, a P value $< 1.0 \times 10^{-3}$ was chosen to give a false discovery rate of $<5\%$. Only reads that aligned to a unique genomic position with no more than two mismatches were retained for the above analysis. When multiple reads mapped to the same position in the genome, only one read was counted. Representative ChIP-seq enriched regions were visualized in the Integrated Genome Browser. To assign ChIP-seq enriched regions (peaks) to genes, we employed Cis-regulatory Elements Annotation System (CEAS) to create average profiling of all Refseq genes

and overlaps of significant peaks with genomic annotation regions. Genes with significant peaks within 5 kb of their TSSs were considered as bound.

**Statistical analysis.** All data were presented as the mean ± s.d. Two-tailed Student's t-test was performed between two groups and a difference was considered statistically significant with $P < 0.05$. The Wilcoxon signed rank test was used to explore the significant differences in IHC staining intensity between different groups. All statistical analyses were performed using the SPSS 17.0 software.

**Data availability.** The authors declare that all relevant data are available within the article and its Supplementary information files or from the corresponding author upon reasonable request. The ChIP-seq datasets have been submitted to the NCBI database under the accession number GSE71885.

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

### Acknowledgements

This work was supported by NIH grants R3713848 and DE15964, and the Shapiro Family Charitable Foundation. We would also like to thank Dr Yi Zhang for the generous gift of KDM3A plasmid and Drs Lee Ellis and Fan Fan for the human colorectal cells.

### Author contributions

The study was conceived and designed by C.-Y.W. and J.L. The TMA staining and statistical analysis were performed by Q.Z. ChIP-seq data analysis was performed by Y.Y. and B.Y. The apoptosis assays were performed by Y.C., S.R. and J.L. Mouse xenografts were generated by J.L. and P.D. All other experiments were performed by J.L. The manuscript was written by J.L. and C.-Y.W.

### Additional information

**Competing interests:** The authors declare no competing financial interests.

