## [Peer Review File · Nature Communications]

Reviewers' comments:

Reviewer #1 (Remarks to the Author):

In their manuscript, KDM3 Epigenetically Controls Tumorigenic Potentials of Human Colorectal Cancer Stem Cells Through Wnt/ β -catenin, Jiong Li et al. demonstrate that histone demethylases of the KDM3 family activates Wnt signaling in CRC cells by erasing H3K9me2 marks on Wnt target genes and recruiting MLL1 to promote H3K4 methylation. Thereby, their data provide insights in the epigenetic control mechanisms of Wnt signaling in CRC cells. The mechanistic insights in the epigenetic control of Wnt signaling in CRC cells is novel and interesting. However, the manuscript in general requires some more expansion and confirmation of the results and interpretational adjustments.

Specific comments:

1) Until now reliable CSC markers are not available and studies to CSC are dependent on functional experimental strategies like limiting dilution assays in order to study the clonogenic potential of cancer cells, one of the hallmarks of CSC. Here, the authors studied tumor formation and growth of control vs. KDM3A/B depleted cells in vivo but the clonogenic potential has not been determined. In order to more formally prove that the CSC activity is impaired of CRC cells upon KDM3a/b knockdown a limiting dilution assay would be required for the in vitro and in vivo experiments. In addition, the authors investigated tumor formation and growth of not necessarily only the CSC population. Even though solely the ALDH+ population is injected into mice, this does not mean that all these cells are CSCs as the CSC state has been described to be subjected to plasticity. Therefore, as well in the title as in the main text of the manuscript, the authors should emphasize more generally the impaired tumorigenic potential of CRC cells instead of stressing the impaired tumorigenic potential of CSCs.

2) Throughout the manuscript some data has been described but is not shown by the authors, e.g. the co-immunoprecipitation of β -catenin and KDM3A/B and the recruitment of KDM3A/B to Wnt target genes. This data should be showed.

3) It is unclear why KDM3A/B knock-down already significantly induced apoptosis in the untreated HCT116 and PS1 cells compared to the control cells. Is this significant compared to untreated scramble cells? The authors should provide an explanation for this observation.

4) The legends are poorly described. This will need to be improved. Also, the authors should declare in the legends whether the data shown are derived from independent experiments

Minor considerations:

1) Based on the siRNA screening of different histone demethylases in figure 1a the authors decided to focus on the role of the KDM3 family members in the inhibition of Wnt signaling. They do not provide a clear explanation why they have chosen to further study the impact of KDM3 on Wnt signaling instead of one of the other histone demethylase families which also seem to inhibit Wnt signaling.

2) It is unclear on which cell line(s) the CHIP-seq experiment has been performed.

3) In figure 3H the authors do not show the CD133-/EPCAM+ as control. This should be added in order to show convincingly that the non- vs tumor-initiating cells are separated in this setting.

4) Throughout the manuscript there are multiple sentences which lack some words. Also, some statements lack a reference. (e.g. first sentence in the section 'KDM3 promote CSCs resistance to apoptosis'.) These should be added.

5) Figure 1m needs to be presented more clear.

6) Overall the figure and supplementary figure sections are not very well balanced. In my opinion some experiments should go to the supplementary section for the sake of clarity. For example similar experiments are performed in different cell lines; e.g. in figure 2a-e and 2f-i. I would suggest to move the validation experiments to the supplementary figures. The same accounts for

Fig 5 G-I; also please provide the zoom in for the WRE in Fig 5g,I and k.

7) Are figure 2a and 2f representative for multiple independent experiments? This is not clear.

Reviewer #2 (Remarks to the Author):

In this manuscript Li and colleagues investigate the potential epigenetic drivers underlying colon cancer stem cells behavior. Using a WNT-reporter screen, they identify several histone demethylases as potential modulators of WNT signalling. They nicely confirm that KDM3A-B control a large subset of genes involved in the WNT signalling cascade. Next the authors investigate the link between KDM3A/B and cancer stem cells in colon cancer models. The data show that inhibition of these KDMs correlate with a reduction in putative CSC cells. Next they investigate the role of KDMs in drug resistance, showing that depletion of these KDMs correlate with increase sensitivity to chemo-treatments. Finally, they assess the potential mechanism driven by KDMs using genome-wide analysis of the histone marks mediated by KDM3B. They show that depletion of KDM3B corresponds to an increase of the repressive mark H3K9me3, suggesting that KDMs act by removing repressive marks at the promoter of WNT target genes. They also present evidence about potential coupling of KDM with MLL protein to combinatorially alter the epigenetic states of WNT target genes. This might occur through removal of H3K9me3 to favour deposition of H3K4m3.

The manuscript is interesting and present novel and original data regarding the underlying mechanisms of WNT activation. The epigenetic analysis is quite compelling but require major clarifications. However, this reviewer is less convinced about the connection between KDM and cancer stem cells. The conclusions from this section should be significantly toned down or more work needs to be carried on to support the current conclusion that KDMs are important to maintain CSC identity thus contributing to drug resistance.

General comments:

- Many of the figure are too crowded and hard to follow. There are too many panels, the panels are too small to be assessed and flow suffers a lot. It would be best to streamline the figures and move some of the validation experiments in the supplementary section.
- It appears the manuscript was put together in quite an hurry. Several sentences are left hanging. For example page 16 " Our results suggest that --???? might act in concert with histone methyltransferases to orchestrate Wnt target gene activation'. Page 18 "These findings highlight that the methylation state of H3K9 would be an important mark indicating the transcriptional status of a given--????". Our ChIP-seq and ChIP results suggested that both KDM3A and KDM3B were...". Page 20 "Our results suggest that targeting ---??? might help...." Many other examples throughout the text.
- The authors only use KDM depletion experiments to investigate epigenetic changes. However, in the initial reporter assay, overexpression of KDM was also used as a more appropriate tool. Many of the results should be better described to highlight these experimental limitations.

Major Experimental Comments.

The manuscript can be divided in two parts. One linking KDM to CSC biology and one linking KDM to epigenetic activation of WNT signaling.
CSC section.

-The authors imply that KDM control CSC behavior and KDMs depletion has a stronger impact on CSC survival. However, there are no data to support this as generally, the effect of siKDMs on

ALDH- cells are not shown.

-It is not clear if siKDM induces differentiation of ALDH+ to ALDH-, selective death of ALDH+ or general cytotoxicity. Can the author clarify?

-How stable is the ALDH+ population? Several studies in other models suggest that CSC tend to spontaneously differentiate rather quickly. How long after sorting were the siRNA experiments conducted?

-Are there other functional marker for CSCs in colon cancer? One might suggest that KDM inhibition causes H3K9 accumulation on the ALDH gene promoter thus inducing its silencing it. (see later comments on off-target effects). This mechanism would be completely independent from WNT inhibition. Can the author rule this out?

- Are DKK1/AXIN2/MYC differentially expressed between ALDH+ and ALDH-?

-Figure 4. I am not convinced that siKDM sensitize cells to chemotherapy. Considering that siRNA doubles the amount of cell death on its own, the ration between basal and chemo-treated should be considered in the analysis. More importantly, the assumption here is that CSC are more chemoresistant than ALDH-. Can the author show the death rate for ALDH- cells treated or not with chemotherapy? From figure 4A-B and 4E-F does not appear that ALDH+ cells are chemo-resistant at all.

Epigenetic section.

Overall, the ChIP-qPCR data/Co-IP are quite convincing. However, there is some serious concerns on the generalizability of the findings.

-The CHIP-seq tracks seem to suggest significant off-target effects. For example, Figure 5E. There is a dramatic increase in H3K9me3 over the AXIN2 genes in correspondence to no KDM3B binding at all. I understand that the author focuses on promoter regions, however these data suggest that siKDM generally invoke a wide-spread increase for the H3K9me3 repressive mark, even in the absence of active KDM binding. How do the authors interpret or control for this? Can the author check with ChIP-qPCR if H3K9me3 is acquired also in these regions?

-It would be better if the authors could try to perform some over-expressing experiments, at least in few focused instances. Even better would be to perform rescue experiments.

-Finally, were the CHIP-seq data obtained from the whole population or from ALDH+ sorted cells? From the other figures someone would expect that KDM3B phenotype is very specific to ALDH+/CSC cells. However, all 90% of the CHIP-seq signal might come from ALDH- cells. For these reason I struggle to reconcile the first part of the manuscript with the latter. It is essential that the author resolves these contradiction prior to publication.

Statement such as page 15 "Taken together, KDM3A/B epigenetically regulate activation of Wnt target genes by directly erasing H3K9me2 marks" are factually incorrect at this stage as there are likely significant secondary effect from KDM silencing. Similarly, statement linking KDM activity to CSC phenotype should be significantly toned down in light of CHIP-seq results.

Minor comments

-What happens to % of ALDH+ cells after upon KDM overexpression? And after the siRNA stops working.

-Please remove data not shown results or provide the data.

-The authors are using HCT-116 (MSI-high) for all of experiments and SW480 (MSS) for some. Thus their data was not dependent on MSI status.

-SF1. Contrary to the KDM3B, KDM3A was mainly stained on cytoplasm. What is the reason of the different localization between KDM3A and KDM3B? Is there any difference of KDM3A and/or KDM3B staining according to the histology grade of CRC?

-The author did Chip PCR in CSCs isolated from cell lines and tumor xenografts. Please clarify the CSC marker which were used?

We thank the reviewers for carefully reading through our manuscript and providing constructive comments which have dramatically helped us to improve our manuscript.

Reviewer 1:

Specific comments:

1) Until now reliable CSC markers are not available and studies to CSC are dependent on functional experimental strategies like limiting dilution assays in order to study the clonogenic potential of cancer cells, one of the hallmarks of CSC. Here, the authors studied tumor formation and growth of control vs. KDM3A/B depleted cells in vivo but the clonogenic potential has not been determined. In order to more formally prove that the CSC activity is impaired of CRC cells upon KDM3a/b knockdown a limiting dilution assay would be required for the in vitro and in vivo experiments. In addition, the authors investigated tumor formation and growth of not necessarily only the CSC population. Even though solely the ALDH⁺ population is injected into mice, this does not mean that all these cells are CSCs as the CSC state has been described to be subjected to plasticity. Therefore, as well in the title as in the main text of the manuscript, the authors should emphasize more generally the impaired tumorigenic potential of CRC cells instead of stressing the impaired tumorigenic potential of CSCs.

As Reviewer 1 suggested, we performed limiting dilution assays and confirmed the clonogenic potential of the ALDH⁺ cells (Supplementary Figure 3c). Additionally, we showed that KDM3A/B knockdown impaired the tumorsphere formation ability of ALDH⁺ cells. We have also changed the title of manuscript as recommended.

2) Throughout the manuscript some data has been described but is not shown by the authors, e.g. the coimmunoprecipitation of β catenin and KDM3A/B and the recruitment of KDM3A/B to Wnt target genes. This data should be showed.

The interaction between β -catenin and KDM3A/B was determined by Co-IP and GST-pull down assays. We have included these data in the new Figure 2. Moreover, our ChIP-seq and ChIP-qPCR experiments revealed that KDM3A/B were recruited to WNT target gene promoters.

3) It is unclear why KDM3A/B knockdown already significantly induced apoptosis in the untreated HCT116 and PS1 cells compared to the control cells. Is this significant compared to untreated scramble cells? The authors should provide an explanation for this observation.

We did observe that KDM3/B knockdown increased the basal level of apoptosis in ALDH⁺ cells isolated from HCT116 and PS1 cells. It is well-known that hyperactivation of Wnt signaling promotes human colorectal cancer cell survival. Since KDM3A/B regulates Wnt/ β -catenin-mediated transcription, it is not surprising that KDM3A/B knockdown increased apoptosis in HCT116 and PS1 cells.

4) The legends are poorly described. This will need to be improved. Also, the authors should declare in the legends whether the data shown are derived from independent experiments.

We have carefully edited the legends. The number of independent experiments is now declared in the legends

Minor considerations:

1) Based on the siRNA screening of different histone demethylases in figure 1a the authors decided to focus on the role of the KDM3 family members in the inhibition of Wnt signaling. They do not provide a clear explanation why they have chosen to further study

the impact of KDM3 on Wnt signaling instead of one of the other histone demethylase families which also seem to inhibit Wnt signaling.

It was based on our educational speculation. H3K9 methylation represses gene transcription. Because KDM3 family members are well-known demethylases for erasing H3K9me2, we picked them the first to characterize their role in Wnt signaling. We are going to examine other histone demethylases in Wnt signaling after this current work is finished.

2) *It is unclear on which cell line(s) the CHIPseq experiment has been performed.*

We used HCT116 cells to perform the ChIP-seq experiment.

3) *In figure 3H the authors do not show the CD133-/EPCAM+ as control. This should be added in order to show convincingly that the non vs tumor initiating cells are separated in this setting.*

We have included the CD133⁻/EPCAM⁺ as control.

4) *Throughout the manuscript there are multiple sentences which lack some words. Also, some statements lack a reference. (e.g. first sentence in the section 'KDM3 promote CSCs resistance to apoptosis'.) These should be added.*

We have revised our manuscript as suggested.

5) *Figure 1m needs to be presented more clear.*

We have included a more detailed description of GSEA assays in the section of Material and Methods.

6) *Overall the figure and supplementary figure sections are not very well balanced. In my opinion some experiments should go to the supplementary section for the sake of clarity. For example similar experiments are performed in different cell lines; e.g. in figure 2ae and 2fi. I would suggest to move the validation experiments to the supplementary figures. The same accounts for Fig 5 Gf; also please provide the zoom in for the WRE in Fig 5g,l and k.*

We have re-organized the figure and supplementary figure sections as suggested.

7) *Are figure 2a and 2f representative for multiple independent experiments? This is not clear.*

Yes, the original Fig. 2a and 2f (supplementary Fig. 2a,d) are representative for one of three independent experiments.

Reviewer 2.

General comments:

Many of the figure are too crowded and hard to follow. There are too many panels, the panels are too small to be assessed and flow suffers a lot. It would be best to streamline the figures and move some of the validation experiments in the supplementary section. It appears the manuscript was put together in quite an hurry. Several sentences are left hanging. For example page 16 " Our results suggest that ???? might act in concert with histone methyltransferases to orchestrate Wnt target gene activation'. Page 18 "These findings highlight that the methylation state of H3K9 would be an important mark indicating the transcriptional status of a given?????. Our ChIPseq and ChIP results suggested that both KDM3A and KDM3B were...". Page 20 "Our results suggest that targeting ??? might help...." Many other examples throughout the text.

As Reviewer 2 suggested, we have streamlined the figures and move some of the validation results in the supplementary section. We have also carefully edited our manuscript.

The authors only use KDM depletion experiments to investigate epigenetic changes. However, in the initial reporter assay, overexpression of KDM was also used as a more appropriate tool. Many of the results should be better described to highlight these experimental limitations.

We performed the initial screening using siRNA since the siRNA of histone demethylases are commercially available. Overexpression of histone demethylases could be used as an alternative tool besides siRNA screening. Unfortunately, we were unable to obtain all cDNA expression vectors of these histone demethylases. However, as Reviewer suggested, we have overexpressed KDM3A/B in multiple experiments to confirm our findings in our revised manuscript (Fig. 1f-J, Fig. 2, Supplementary Fig.2b and 6e-h).

Major Experimental Comments.

CSC section.

The authors imply that KDMs control CSC behavior and KDMs depletion has a stronger impact on CSC survival. However, there are no data to support this as generally, the effect of siKDMs on ALDH⁻ cells are not shown.

We have included data that showed the effect of KDM3A/B siRNA on ALDH⁻ cells as requested. The ALDH⁺ cells were more resistant to apoptosis induced by both cisplatin and irinotecan (Supplementary Figure 4).

It is not clear if siKDM induces differentiation of ALDH⁺ to ALDH⁻, selective death of ALDH⁺ or general cytotoxicity. Can the author clarify? How stable is the ALDH⁺ population? Several studies in other models suggest that CSC tend to spontaneously differentiate rather quickly. How long after sorting were the siRNA experiments conducted?

To maintain the maximum cancer stem cell-like properties of the ALDH⁺ cells, siRNA were transfected immediately after sorting and experiments were immediately performed 24 hr after transfection.

Are there other functional markers for CSCs in colon cancer? One might suggest that KDM inhibition causes H3K9 accumulation on the ALDH gene promoter thus inducing its silencing it. (see later comments on off target effects). This mechanism would be completely independent from WNT inhibition. Can the author rule this out?

According to our ChIP-seq data, there were no peaks of KDM3B on the *ALDH1A1* gene locus including the promoter area, and knockdown of KDM3A/B did not affect H3K9me2 levels around the *ALDH1A1* promoter (Supplementary Fig. 9b). CD44v6 has been reported as a functional marker for CSCs in CRC. We isolated CD44v6⁺/EpCAM⁺ cell populations from xenografts. The CD44v6⁺/EpCAM⁺ cells effectively formed sphere-like colonies *in vitro*. Knockdown of KDM3A/B dramatically inhibited the tumorsphere formation ability of these cells (Supplementary Fig. 3a). These results ruled out the possibility that KDM inhibition caused H3K9me accumulation on the ALDH gene promoter thus inducing gene silencing.

Are DKK1/AXIN2/MYC differentially expressed between ALDH⁺ and ALDH⁻?

The expression of AXIN2, DKK1 and MYC were higher in PS1-ALDH⁺ cells as compared to PS1-ALDH⁻ cells. We have included these data in Fig. 4f and i.

Figure 4. I am not convinced that siKDM sensitize cells to chemotherapy. Considering that siRNA doubles the amount of cell death on its own, the ration between basal and chemotreated should be considered in the analysis. More importantly, the assumption here is that CSC are more chemoresistant than ALDH. Can the author show the death rate for ALDH- cells treated or not with chemotherapy? From figure 4AB and 4EF does not appear that ALDH+ cells are chemoresistant at all.

As described above, the depletion of KDM3A/B induces the differentiation of ALDH⁺ to ALDH⁻ and it was not surprising that KDM3A/B siRNA also induced mild apoptosis of ALDH⁺ cells. However, this mild induction of apoptosis was much lower than apoptosis induced by cisplatin and irinotecan (Fig. 5). Since the cell death was triggered by different initiators, we believe that it was more appropriate to compare the percentage of cell death rather than the ratio to determine the chemo-resistant effect. We have also included the effect of KDM3A/B siRNA on ALDH⁻ cells as requested. The ALDH⁺ cells were more resistant to apoptosis induced by both cisplatin and irinotecan (Supplementary Fig. 4).

Epigenetic section.

Overall, the ChIPqPCR data/CoIP are quite convincing. However, there is some serious concerns on the generalizability of the findings.

The CHIPseq tracks seem to suggest significant off target effects. For example, Figure 5E. There is a dramatic increase in H3K9me2 over the AXIN2 genes in correspondence to no KDM3B binding at all. I understand that the author focuses on promoter regions, however these data suggest that siKDM generally invoke a widespread increase for the H3K9me2 repressive mark, even in the absence of active KDM binding. How do the authors interpret or control for this? Can the author check with ChIPqPCR if H3K9me2 is acquired also in these regions?

We previously focused on the promoter region of *AXIN2*. However, ChIP-seq indeed identified that there was another KDM3B binding peak located in the third intron region of the *AXIN2* gene. The recruitment of KDM3B to this locus was confirmed by ChIP-qPCR (Supplementary Fig. 6a, A4 site for qPCR validation). KDM3A/B depletion resulted in a significant reduction of KDM3B recruitment to this region and together with enhanced H3K9me2 enrichment in its upstream (A3) and downstream regions (A5) respectively, which were validated by qPCR. More importantly, we have also identified seven TCF binding sites within 1kb of this KDM3B binding site. ChIP-qPCR suggested that β -catenin could also be recruited to this site as compared with negative control region. Finally, we also overexpressed HA-KDM3B in HCT116 cells and found that HA-KDM3B could be detected on this site with ChIP-qPCR using anti-KDM3B or anti-HA antibodies (Supplementary Fig. 6e-h). Based on our ChIP-seq results, KDM3B peaks reduction and H3K9me2 peaks induction were quite overlapped on the well-characterized WNT target genes *AXIN2*, *DKK1*, *CCND1* and *MYC*. Furthermore, gene ontology analysis using genes associated with upregulated H3K9me2 enrichment clearly suggested that WNT signaling is the most related signaling pathway affected by KDM3A/B depletion. In summary, our data strongly suggest that KDM3 interact with β -catenin and specifically regulate WNT target gene transcription.

It would be better if the authors could try to perform some overexpressing experiments, at least in few focused instances. Even better would be to perform rescue experiments.

We stably overexpressed HA-KDM3B in HCT116 cells. We found that there was a significant induction of KDM3B recruitments to WNT target gene promoters (WRE) including *AXIN2*, *DKK1* and *CCND1*. Overexpression of KDM3B could also decrease the H3K9me2 level of these regions (Supplementary Fig. 6e-h).

Finally, were the CHIPseq data obtained from the whole population or from ALDH+ sorted cells? From the other figures someone would expect that KDM3B phenotype is very specific to ALDH+/CSC cells. However, all 90% of the CHIP seq signal might come from ALDH- cells. For these reason I struggle to reconcile the first part of the manuscript with the latter. It is essential that the author resolves these contradiction prior to publication.

The ChIP-seq experiments were performed in whole population of HCT116 cells, not from ALDH⁺ sorted cells. Since ChIP-seq requires large amount of cells, it was very difficult for us to obtain enough ALDH⁺ cells to perform the experiments. HCT116 cells have an APC mutation, and β -catenin is constitutively activated in both ALDH⁺ and ALDH⁻ cells. However, β -catenin/Tcf-mediated transcription was higher in ALDH⁺ cells than in ALDH⁻ cells. Therefore, our strategy was to perform ChIP-seq in HCT116 cells and then test our findings in ALDH⁺ cells by individual ChIP-qPCR. Our ChIP-qPCR has confirmed the critical role of KDM3A/B in WNT signaling in HCP1-ALDH⁺ cells (Fig. 7). As Reviewer suggested, we have revised our manuscript accordingly.

Statement such as page 15 "Taken together, KDM3A/B epigenetically regulate activation of Wnt target genes by directly erasing H3K9me2 marks" are factually incorrect at this stage as there are likely significant secondary effect from KDM silencing. Similarly, statement linking KDM activity to CSC phenotype should be significantly toned down in light of CHIPseq results.

As Reviewer suggested, we revised our statements and toned down our conclusion. Additionally, we have provided additional data to support our findings.

Minor comments

What happens to % of ALDH+ cells after upon KDM overexpression? And after the siRNA stops working.

Overexpression of KDM3A and KDM3B increased the ALDH⁺ population from 10.7% to 16.3% (Supplementary Fig. 2b). We found that the ALDH⁺ population back to normal after 10 days of siRNA transfection (Supplementary Fig. 9a).

Please remove data not shown results or provide the data.

We included the data about the interaction between β -catenin and KDM3A/B that was not shown in the original manuscript.

The authors are using HCT116 (MSI high) for all of experiments and SW480 (MSS) for some. Thus their data was not dependent on MSI status.

Thanks for this comments. In this study we did not focus on microsatellite instability.

SF1.Contrary to the KDM3B, KDM3A was mainly stained on cytoplasm. What is the reason of the different localization between KDM3A and KDM3B? Is there any difference of KDM3A and/or KDM3B staining according to the histology grade of CRC?

Cancer cells in tumors are not homogenously equal. For KDM3A staining, over 50% of cells represented robust nuclear staining. KDM3A has also been found to act as a Hsp90 client protein that interacts with Hsp90 in the cytoplasm and participates directly in the regulation of cytoskeletal components in the elongation process of sperm head in mice (Kasioulis et al., Mol Biol Cell. 2014, 25:1216-33). Therefore, the localization of KDM3A in cytoplasm could be explained by the association with Hsp90 in cytoplasm. To further confirm our results, we isolated the nuclear and cytoplasmic extract from HCT116 and SW480 cells. Western blot showed that a considerable amount of KDM3A could be detected in cytoplasmic extract of both

cell lines. However, KDM3B appeared to be largely present in nucleus as compared with KDM3A (Supplementary Fig. 9c). We performed the correlation analysis between tumor grade and tumor stage with the abundance of KDM3A/B. No significant correlation was found.

The author did Chip PCR in CSCs isolated from cell lines and tumor xenografts. Please clarify the CSC marker which were used?

We have specified the CSC makers which were used in our experiments.

Reviewers' comments:

Reviewer #1 (Remarks to the Author):

Basically all of my scientific comments have been addressed either with new data or important clarifications. The presented data provides an important advance in the field. However, I still feel the manuscript is somewhat sloppy compiled and written. This should be meticulously checked again before the paper can be considered for publication.

Reviewer #2 (Remarks to the Author):

The authors have attempted to resolve some of the technical/biological issues and I appreciate their effort. There is still a critical question that they have carefully avoided even if they have managed to create the correct model:

-They now have an overexpression KDM3a system that seems to be working correctly. Prior to accept the manuscript I think it is essential they perform some phenotypical assays (rescue for example) to see they can revert the phenotype they got via siRNA. A simple start would be, are the KDM3a o.e. cells more resistant to chemotherapy?

We thank the reviewers for carefully reading through our manuscript and providing constructive comments which have dramatically helped us to improve our manuscript.

Reviewer 1

No concerns. Thanks.

Reviewer 2.

“They now have an overexpression KDM3a system that seems to be working correctly. Prior to accept the manuscript I think is essential they perform some phenotypical assays (rescue for example) to see they can revert the phenotype they got via siRNA. A simple start would be, are the KDM3a o.e. cells more resistant to chemotherapy?”

As Reviewer 2 suggested, we overexpressed KDM3B in ALDH⁺HCT116 cells and KDM3A/B knockdown ALDH⁺HCT116 cells (Supplementary Fig. 4c). We found that overexpression of KDM4B promoted ALDH⁺HCT116 cell survival. The restoration of KDM4B expression in KDM3A/B knockdown ALDH⁺HCT116 cells also significantly restored cell resistance to cisplatin and irinotecan (Supplementary Fig. 4d).